＃ nature communications

# Local molecular and global connectomic contributions to cross-disorder cortical abnormalities

Justine Y. Hansen [1], Golia Shafiei [1], Jacob W. Vogel [2], Kelly Smart [3], Carrie E. Bearden [4], Martine Hoogman[5,6], Barbara Franke[5,6], Daan van Rooij[6], Jan Buitelaar [6], Carrie R. McDonald[7], Sanjay M. Sisodiya[8], Lianne Schmaal [9], Dick J. Veltman[10], Odile A. van den Heuvel[10,11], Dan J. Stein [12], Theo G. M. van Erp [13], Christopher R. K. Ching[14], Ole A. Andreassen[15], Tomas Hajek[16], Nils Opel[17], Gemma Modinos[18], André Aleman[19], Ysbrand van der Werf [11], Neda Jahanshad[14], Sophia I. Thomopoulos[14], Paul M. Thompson[14], Richard E. Carson [3], Alain Dagher [1] & Bratislav Misic [1] ✉

Numerous brain disorders demonstrate structural brain abnormalities, which are thought to arise from molecular perturbations or connectome miswiring. The unique and shared contributions of these molecular and connectomic vulnerabilities to brain disorders remain unknown, and has yet to be studied in a single multi-disorder framework. Using MRI morphometry from the ENIGMA consortium, we construct maps of cortical abnormalities for thirteen neurodevelopmental, neurological, and psychiatric disorders from $N = 21,000$ participants and $N = 26,000$ controls, collected using a harmonised processing protocol. We systematically compare cortical maps to multiple micro-architectural measures, including gene expression, neurotransmitter density, metabolism, and myelination (molecular vulnerability), as well as global connectomic measures including number of connections, centrality, and connection diversity (connectomic vulnerability). We find a relationship between molecular vulnerability and white-matter architecture that drives cortical disorder profiles. Local attributes, particularly neurotransmitter receptor profiles, constitute the best predictors of both disorder-specific cortical morphology and cross-disorder similarity. Finally, we find that cross-disorder abnormalities are consistently subtended by a small subset of network epicentres in bilateral sensory-motor, inferior temporal lobe, precuneus, and superior parietal cortex. Collectively, our results highlight how local molecular attributes and global connectivity jointly shape cross-disorder cortical abnormalities.

---

The brain is a network with intricate connection patterns among individual neurons, neuronal populations, and large-scale brain regions. The wiring of the network supports propagation of electrical signals, as well as molecules needed for growth and repair. This complex system is vulnerable to multiple neurological, psychiatric and neurodevelopmental disorders. Pathological perturbations—including altered cellular morphology, cell death, aberrant synaptic pruning and miswiring—disrupt inter-regional communication and manifest as overlapping groups of sensory, motor, cognitive and affective symptoms. How different disorders are shaped by local and global vulnerability is unknown.

Indeed, several studies have demonstrated cross-disorder connectomic vulnerability, where regions and white-matter pathways are targeted non-randomly. In particular, regions that are highly connected and potentially important for communication tend to be disproportionately affected by disease[1,2]. A similar phenomenon is observed for connections that support multiple communication pathways[3]. In neurodegenerative diseases such as Alzheimer's and Parkinson's diseases, emerging evidence suggests pathological misfolded proteins spread trans-synaptically, such that the connectivity of the brain shapes the course and expression of these diseases[4-11]. Recent evidence also suggests that patterns of tissue volume loss in schizophrenia are circumscribed by structural and functional connection patterns[12,13]. Collectively, these studies demonstrate that both neurodevelopmental and neurodegenerative brain diseases are influenced by network connectivity[14,15].

The effects of disease can also be driven by local cellular and molecular vulnerability. Namely, local patterns of gene expression[16-18], neurotransmitter receptor profiles[19], cellular composition[20], and metabolism[21-24] may predispose individual regions to stress and, ultimately, pathology. Importantly, local and global vulnerability are not necessarily mutually exclusive; some diseases may originate from local pathologies that spread selectively along the network to other vulnerable regions. How local attributes and global connectivity shape cross-disorder pathology remains an open question.

Here, we map local molecular attributes ("molecular vulnerability") and global network connectivity ("connectomic vulnerability") to case versus control cortical thickness abnormalities of thirteen different neurological, psychiatric, and neurodevelopmental diseases and disorders from the ENIGMA consortium[25]. We consistently find that disorder-specific cortical abnormality is shaped more by the local molecular fingerprints of brain regions than network embedding. Interestingly, for disorders that are better predicted by molecular attributes, we find that the spatial patterning of cortical abnormalities reflects the underlying network architecture, suggesting that the joint contribution of local molecular and global connectomic mechanisms is greater than their individual contribution. Next, we study cross-disorder similarity and find that regions with similar molecular make-up tend to be similarly affected across disorders. Collectively, the present report highlights how local and global factors interact to shape cross-disorder cortical morphology.

## Results

We collected thirteen spatial maps of cortical abnormalities from the ENIGMA consortium for the following diseases, disorders, and conditions: 22q11.2 deletion syndrome[26], attention-deficit/hyperactivity disorder (ADHD)[27], autism spectrum disorder (ASD)[28], idiopathic generalised epilepsy[29], right temporal lobe epilepsy[29], left temporal lobe epilepsy[29], depression[30], obsessive-compulsive disorder (OCD)[31], schizophrenia[32], bipolar disorder[33], obesity[34], schizotypy[35], and Parkinson's disease[36]. For simplicity, we refer to diseases, disorders, and conditions as "disorders" throughout the text. While most disorders show decreases in cortical thickness, some (e.g., 22q11.2 deletion syndrome, ASD, schizotypy) also show regional increases in cortical thickness. We therefore refer to the cortical measure as "cortical

abnormality". All cortical abnormality maps were collected from adult participants (except ASD which included younger participants), following identical processing protocols, for a total of over 21,000 scanned participants against almost 26,000 controls. To assess the extent to which each abnormality pattern is informed by molecular attributes and network connectivity, we defined a molecular and connectivity fingerprint at each brain region. The molecular fingerprint of a region was defined using the gene expression gradient (a potential proxy for cell type distribution[16,20,37-39]), neurotransmitter receptor gradient, excitatory-inhibitory receptor density ratio, glycolytic index, glucose metabolism, synapse density, and myelination (Fig. 1a). Likewise, we defined the connectivity fingerprint of a region by calculating the strength, betweenness centrality, closeness centrality, mean Euclidean distance, participation coefficient, clustering coefficient, and mean first passage time of a weighted structural connectivity matrix from 70 healthy adults (Fig. 1b; see *Methods* for details). Collectively, these graph measures aim to capture the connectedness, centrality, and connection diversity of regions in the network. All analyses were conducted using the 68-region Desikan-Killiany parcellation[40,41], as this is the native and only available representation of ENIGMA datasets.

### Local and global contributions to disorder-specific cortical morphology

To assess the extent to which cortical abnormalities of all thirteen disorders are informed by molecular gradients versus measures of network connectivity, we fit a multilinear model between molecular or connectivity predictors and abnormality maps for each disorder separately, for a total of 13 × 2 = 26 model fits (Fig. 2a; for results when molecular and connectomic predictors are combined, see Supplementary Fig. 1). Next, we conducted a dominance analysis for each multilinear model[42-45]. Dominance analysis distributes the $R^2_{adj}$ across input variables as a measure of contribution ("dominance") that each input variable has on the cortical abnormality pattern (Fig. 2b). Each model was cross-validated in a distance-dependent manner (Supplementary Fig. 2[39]).

We find that the fit between molecular predictors and cortical abnormality is greater than that between connectivity predictors and cortical abnormality for most disorders (Fig. 3). Notably, the variance in cortical thickness of schizotypy (a possible precursor of schizophrenia that is poorly defined in the brain[46]) and idiopathic generalised epilepsy (a form of epilepsy that is thought to be informed by genetics instead of brain structural abnormalities[47,48]) are poorly explained by both biological gradients and network measures of the brain. On the other hand, ADHD, ASD, OCD, Parkinson's, and depression are better predicted by molecular predictors, whereas schizophrenia, 22q11.2 deletion syndrome, and bipolar disorder are better predicted by connectivity predictors (Fig. 3).

From the dominance analysis, we find that certain predictors are consistently unimportant. Indeed, synapse density and myelination demonstrate less dominance than microscale gradients such as the gene expression gradient (a potential proxy for cell type distribution[16,20,37-39]), neurotransmitter receptor gradient, and metabolic gradients. Connectivity predictors, particularly measures of centrality, demonstrate less dominance than more fundamental measures of connectivity such as number of connections (strength), distance, and connection diversity (participation coefficient). For completeness, we tested a third family of predictors related to temporal dynamics: magnetoencephalography (MEG)-derived power spectral densities for six canonical frequency bands (Supplementary Fig. 3).

One important consideration with this analysis is that disorder-specific pathology and symptom presentation are heterogeneous over time. The analysis in Fig. 2 is limited to adults and encompasses multiple stages of disease progression. We therefore sought to investigate changes across different ages (paediatric, adolescent, and adult) and

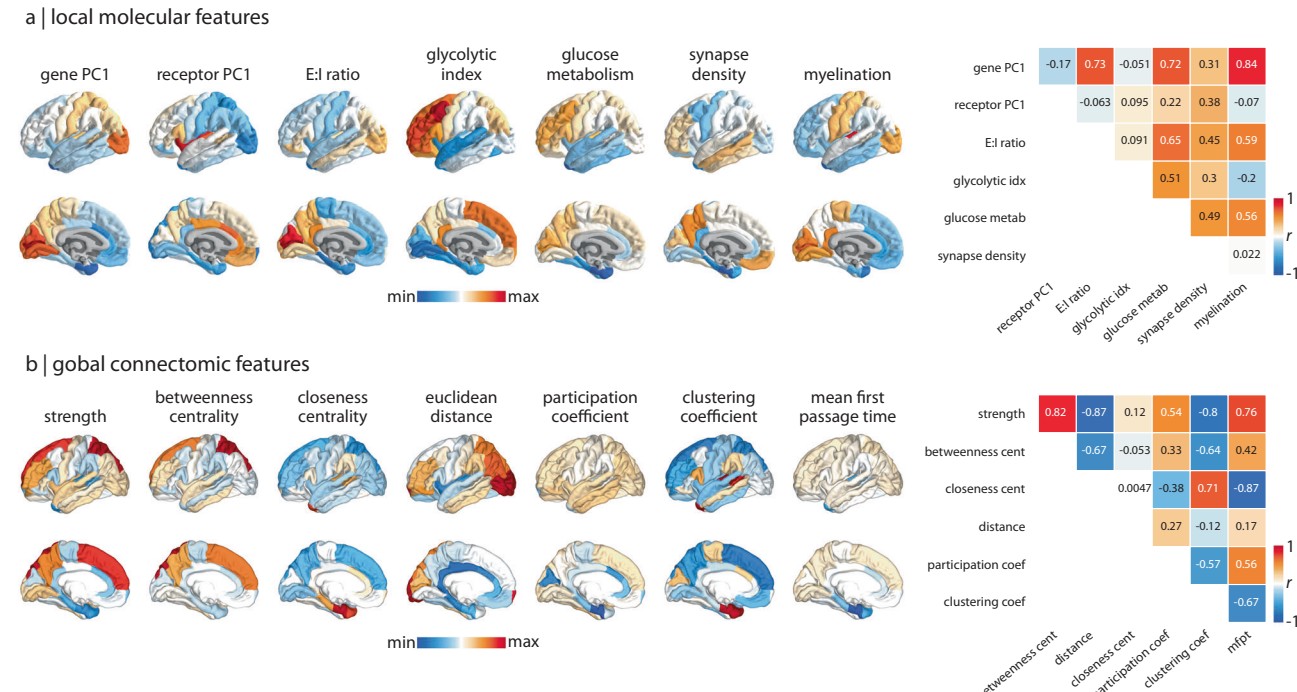

**Fig. 1 | Molecular and connectomic cortical profiles. a, b** Brain surfaces show the z-scored molecular (**a**) and connectomic (**b**) predictors used in the multilinear regression models. Heatmaps on the right show Pearson's correlation coefficients between pairs of features. See *Methods* for details on how each feature was derived. Molecular predictors: gene PC1 = first component of 11 560 genes' expression; receptor PC1 = first component of 18 PET-derived receptor/transporter density; E:I ratio = excitatory:inhibitory receptor density ratio; glycolytic index = amount of aerobic glycolysis; glucose metabolism = [$^{18}$F]-labelled fluorodeoxyglucose (FDG) PET image; synapse density = synaptic vesicle glycoprotein 2A (SV2A)-binding [$^{11}$C]UCB-J PET tracer; myelination = T1w/T2w ratio. Connectivity predictors: strength = sum of weighted connections; betweenness = fraction of all shortest paths traversing region *i*; closeness = mean shortest path length between region *i* and all other regions; Euclidean distance = mean Euclidean distance between region *i* and all other regions; participation coefficient = diversity of connections from region *i* to the seven Yeo-Krienen resting-state networks[164]; clustering = fraction of triangles including region *i*; mean first passage time = average time for a random walker to travel from region *i* to any other region.

different disease severities. First, we tracked the model fit ($R^2_{adj}$) of regression models that fit molecular/connectomic features to paediatric, adolescent, and adult cortical abnormality profiles for the four available disorders with this data (ADHD, bipolar disorder, depression, and OCD; Supplementary Fig. 4a). We find that model fit is greatest in adulthood, except for OCD, which shows little change for connectomic predictors and a lower model fit in adulthood for molecular predictors. Model fit significantly improves when molecular features are used to predict cortical abnormality patterns of ADHD and depression ($F > F_{critical}$, one-sided). Next, focusing on disease severity, we show how model fit changes across four levels of Parkinson's disease severity (Hoehn and Yahr (HY) stages[49]; Supplementary Fig. 4b). Interestingly, we find that from stage HY2, molecular predictors perform worse with disease severity whereas connectomic predictors perform better (although note the changes in model fit are not statistically significant), supporting the notion that Parkinson's pathology is influenced by the spread of misfolded proteins on the structural connectome[10,50,51]. Altogether, these analyses provide a more nuanced and transdiagnostic representation of molecular and connectomic contributions to cortical disorder vulnerability.

**Interactions between local and global vulnerability**
The previous section separately addresses molecular and connectomic contributions to disease-specific cortical abnormalities. However, molecular attributes likely interact with network connectivity to shape disease pathology. These molecular mechanisms include gene expression, neurotransmitter expression, and metabolic pathways in the cell. In neurodegenerative diseases, this interaction may result in synaptic pruning and cortical atrophy whereas in neurodevelopmental disorders, the pathology may manifest as perturbations in network

wiring during development[52]. We hypothesised that abnormalities in such molecular mechanisms at the regional level may spread transsynaptically between connected regions, resulting in connectome-informed changes in cortical morphology that reflect an interplay between local vulnerability and network structure. For instance, two regions may both participate in many connections (have high degree), but one may be connected to more regions with local vulnerability. Thus, despite the fact that their connectomic profiles are similar, one region may have greater disease exposure than the other[10,53].

To test the hypothesis that a region's cortical thickness is driven by "exposure" to abnormalities of connected regions, we measured the extent to which disorders demonstrate network-spreading patterns of cortical morphology[9,13,54]. The extent to which a disorder displays network-informed cortical changes is defined as the correlation between regional abnormality and mean abnormality of structurally connected neighbours (Fig. 4a). Importantly, significance was assessed using the spin-test to control for the effect of spatial autocorrelation on cortical abnormality patterns. We also test the hypothesis that this network-spreading effect is functionally informed, whereby the cortical thickness of structurally connected neighbours is weighted by the functional connectivity between regions when calculating the mean (Fig. 4b; see *Methods* for details and Supplementary Figs. 5 and 6 for scatter plots of regional abnormality versus mean neighbour abnormality across all thirteen disorders). We find that multiple disorders display a significant correlation between regional abnormality and abnormality of connected neighbours ($0.23 < r < 0.80$), suggesting that spatial patterning of disorders reflects the connection patterns between brain regions, above and beyond the effect of spatial autocorrelation (Supplementary Fig. 5).

Does molecular or connectomic predictability of a disorder pattern (Fig. 2a) relate to network spreading? Interestingly, the extent to

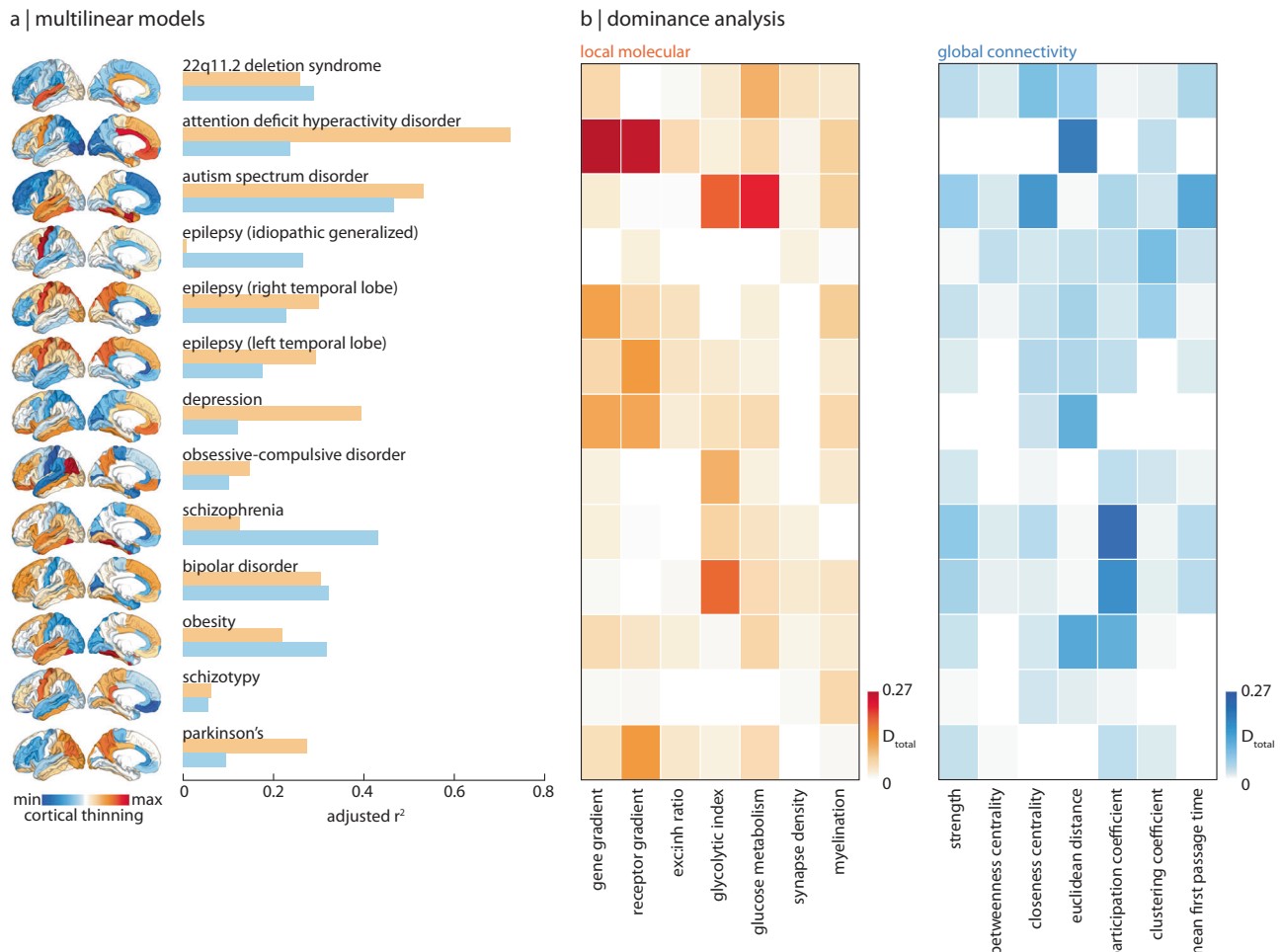

**Fig. 2 | Local and global contributions to disorder-specific cortical morphology. a** A total of twenty-six multilinear models were fit between local molecular and global connectome predictors to cortical abnormality maps of thirteen different disorders (surface plots, left). Adjusted $R^2$ is shown in the bar plot (orange: molecular; blue: connectivity). **b** Dominance analysis was applied to assess the contribution of each input variable (done separately for molecular (orange) and connectivity (blue) predictors) to the fit of the model.

which a disorder can be predicted from molecular attributes (i.e., yellow $R^2$ in Fig. 2a) is positively correlated with the extent to which a disorder displays evidence of network spreading ($r(11) = 0.61$, $p = 0.03$, CI = [0.10, 0.87] when weighted by SC only as shown in Fig. 4a; $r(11) = 0.75$, $p = 0.003$, CI = [0.33, 0.92] when weighted by FC and SC as shown in Fig. 4b). Notably, we do not observe this relationship with the extent to which a disorder can be predicted from global connectivity (i.e., blue $R^2$ in Fig. 2a; $r(11) = 0.24$, $p = 0.42$, CI = [−0.36, 0.70] when weighted by SC only, Fig. 4a; $r(11) = 0.06$, $p = 0.84$, CI = [−0.50, 0.59] when weighted by FC and SC, Fig. 4b). In other words, for disorders with cortical morphologies that more strongly depend on molecular attributes, we also observe a greater effect of disorder exposure. Although we previously found that the cortical patterning of a disorder is less influenced by network embedding per se (e.g., centrality or connection diversity), here we show that it is instead more influenced by network-driven exposure to regions with local vulnerability. This finding is significant because it suggests that the combined effect of local vulnerability and connectome architecture is greater than their individual contribution.

Brain regions with high abnormality and high neighbour abnormality are likely to act as an epicentre of the network-spreading disorder pattern, since the region is both heavily affected and facilitates the spread of atypical morphology[13,53,55]. We calculated epicentre likelihood of each brain region as the mean rank of regional and

neighbour abnormality, such that regions with high node and neighbour abnormality would be labelled as likely epicentres (Fig. 4c). The measure identifies "disorder hubs"—regions that are both vulnerable to disorder-specific changes but also embedded in a highly atypical network cluster. Epicentre likelihood was only calculated for brain maps with significant correlation between their node and neighbour abnormality (network-spreading disorders), and we did not find evidence for epicentre likelihood being driven by distance (Supplementary Fig. 14a. This list comprised of: 22q11.1 deletion syndrome, ADHD, ASD, right and left temporal lobe epilepsy, bipolar disorder, and schizotypy (Supplementary Fig. 7). Next, we aimed to construct a single cross-disorder epicentre likelihood map (Fig. 4c). To avoid having left and right temporal lobe epilepsy—which demonstrate similar epicentre likelihood maps—bias the cross-disorder likelihood map, we combined left and right epilepsy epicentre likelihood into a single average map. We calculated the median epicentre likelihood across these six disorders and find that cross-disorder epicentre likelihood is highest in bilateral sensory-motor cortex, angular gyrus, inferior temporal lobe, precuneus, and superior parietal cortex. In Supplementary Fig. 8 we show mean epicentre likelihood as well as a map that shows the frequency with which a brain region is in the top 50% of most likely epicentres across the six disorders. Across all three methods (mean, median, frequency), cross-disorder epicentre likelihood is consistent.

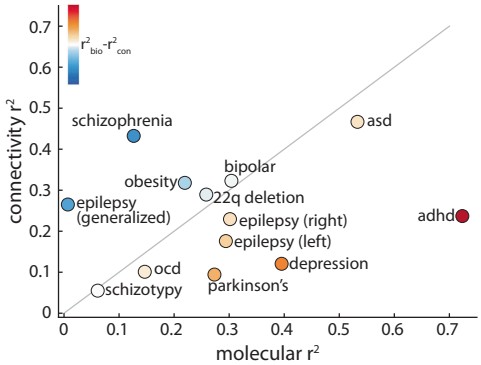

**Fig. 3 | Comparing molecular and connectomic contributions to disorder-specific cortical differences.** The local molecular $R^2_{adj}$ of each disorder is plotted against the global connectivity $R^2_{adj}$. The grey line indicates the identity line and circle colour represents the difference between molecular and connectomic $R^2_{adj}$, such that warm colours represent disorders that are better predicted by molecular predictors, and cool colours represent disorders that are better predicted by connectomic predictors.

## Brain regions with similar molecular annotations are similarly affected across disorders

In the previous sections, we mapped molecular annotations and network measures to each disorder separately. Here, we focused on disorder similarity. For every region we constructed a 13-element vector of abnormality values, where each element corresponds to cortical change (i.e., cortical abnormality) in that region in one disorder. We then correlated regional vectors with each other to estimate how similarly two regions are affected across the thirteen disorders (Fig. 5a). Disorder similarity is analogous to other measures of inter-regional attribute similarity including anatomical covariance[56–58], morphometric similarity[59], correlated gene expression[60–62], receptor similarity[63], temporal profile similarity[44], and microstructural similarity[64].

We first asked whether brain regions with similar molecular versus connectivity fingerprints show greater disorder similarity. Molecular similarity was likewise computed as the pairwise regional correlation of molecular predictors, and vice versa for connectivity. We find that disorder similarity is significantly correlated with molecular similarity ($r(2276) = 0.45$, $p_{spin} = 0.0001$, CI = [0.42, 0.49]; Fig. 5c). On the other hand, the correlation between distance-regressed connectivity similarity and disorder similarity is smaller and non-significant ($r(2276) = 0.25$, $p_{spin} = 0.063$, CI = [0.21, 0.29]; Fig. 5d).

Two of the molecular predictors included in the present report are summary measures of much more expansive molecular annotations: the gene expression gradient and the neurotransmitter receptor gradient. We therefore asked whether inter-regional similarity of these molecular attributes confers similar predisposition to disease. We computed correlated gene expression and neurotransmitter receptor similarity matrices[63,65], and correlated these matrices with disorder similarity (Fig. 5e, f). We find a significant correlation between disorder similarity and neurotransmitter receptor similarity ($r(2276) = 0.41$, $p_{spin} = 0.001$, CI = [0.38, 0.45]), as well as correlated gene expression ($r(559) = 0.46$, $p_{spin} = 0.0001$, CI = [0.40, 0.53])[60,63]. These results hold when distance-regression is applied to the similarity networks instead of spin-tests (Supplementary Fig. 9). Altogether, our results indicate that regions with similar molecular composition are similarly affected across disorders.

We finally ask whether disorder similarity might analogously be informed by structural and functional connectivity between regions. We compared the disorder similarity matrix to weighted structural and functional connectomes. First, we find that brain regions that are structurally connected are more likely to change similarly across

disorders than regions that are not structurally connected, although this result is non-significant against a degree and edge-length preserving null model (Fig. 5g[66]; see *Null models*). Second, we find that brain regions that are within the same intrinsic functional network are more likely to change similarly than regions between functional networks, against the spin-test (Fig. 5g, $p_{spin} = 0.01$). Finally, we find a positive significant correlation between disorder similarity and functional connectivity ($r(2276) = 0.36$, $p_{spin} = 0.004$, CI = [0.33, 0.40]; Fig. 5h). Consistent with the previous subsection, these results collectively suggest that areas that share molecular attributes and connections are similarly affected across disorders.

## Sensitivity and robustness analyses

To ensure the results are not driven by choice of dataset, acquisition parameters and processing methodology, we repeated the analyses using structural and functional networks from the Human Connectome Project ($N = 326$), for which acquisition parameters and processing methodologies differ. The connectomic predictors from the Lausanne dataset used in the main text are highly correlated with the same metrics calculated using HCP data (Supplementary Fig. 10). As a result, the regression models and dominance analyses show consistent results (Supplementary Fig. 11). We also repeat the analysis in Fig. 2 using connectomic predictors calculated based on the binary structural connectome and the functional connectome from the Lausanne dataset (Supplementary Fig. 12).

Next, since the Desikan-Killiany atlas parcellates the brain into unequally sized parcels, we tested the effect of parcel size on disorder abnormality maps. Parcel size was defined as the number of voxels assigned to the parcel using the MNI-152 volumetric parcellation. Across all thirteen disorder maps, we do not find a significant correlation between parcel size and cortical abnormality (Supplementary Fig. 13). Likewise, we compare effects of parcel size on epicentre likelihood maps (Supplementary Fig. 14b). We find no significant correlations except between parcel size and bipolar epicentre likelihood ($r = 0.44$, $p_{spin} = 0.03$). Finally, since epicentre likelihood is calculated using the structural connectome, we also assessed the relationship between epicentre likelihood and distance. Specifically, we correlate epicentre likelihood with the average distance between a brain region and all other brain regions (Supplementary Fig. 14a). We do not find any significant correlations between epicentre likelihood and distance.

## Discussion

In the present report, we comprehensively map local molecular attributes and global measures of connectivity to the cortical morphology of thirteen different neurological, psychiatric, and neurodevelopmental disorders. We consistently find that local attributes govern both disorder-specific abnormalities and cross-disorder similarity more than global connectivity. In addition, we find that molecular mechanisms interact with the structural and functional architectures of the brain to guide cross-disorder abnormality patterns. Altogether, our results highlight how molecular and connectomic vulnerability shape cross-disorder cortical abnormalities.

This work builds on a growing literature on cross-disorder effects, and how shared vulnerability may potentially transcend traditional diagnostic boundaries[3,67–69]. It is becoming increasingly clear that pathology is governed by layers of abnormal processes, at the molecular and cellular level, to neural dynamics, to large-scale brain networks. Aligning high-quality maps of disorder-specific cortical changes to a common reference frame of local and global attributes allows us to systematically relate the effect of disease to multiple scales of organisation. By taking a cross-modal and cross-disorder approach we reveal that, despite different clinical presentation and label, there exists some commonality across diseases including predictors that are ubiquitously important as well as interplay between local vulnerability and network structure.

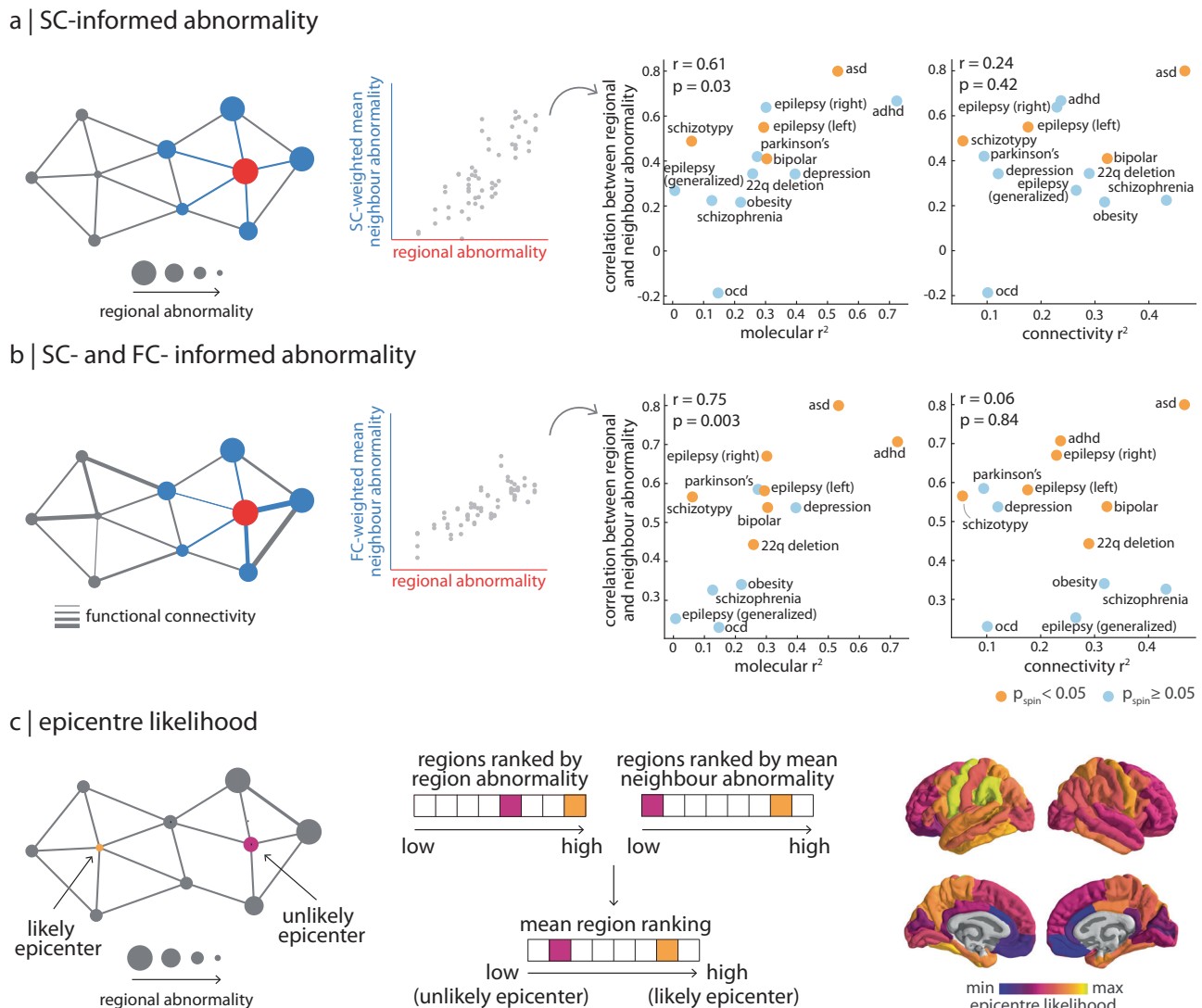

**Fig. 4 | Interactions between molecular and connectomic vulnerability. a** Left: schematic of structural connectivity informing disorder-related cortical changes. The correlation between SC-weighted mean neighbour abnormality and region abnormality represents the extent to which a disorder demonstrates network-spreading disorder-specific cortical morphology. Right: this correlation coefficient was then correlated (Pearson's *r*, two-sided) to both local molecular (left) and global connectivity (right) $R^2_{adj}$. Yellow points refer to disorders where the correlation between region abnormality and SC-weighted mean neighbour abnormality is significant ($p_{spin} < 0.05$). **b** Left: likewise, mean neighbour abnormality can be additionally weighted by functional connectivity between regions. Right: correlation (Pearson's *r*, two-sided) between the extent to which a disorder demonstrates SC- and FC-informed network-spreading cortical morphology and local molecular (left) and global connectivity (right) $R^2_{adj}$. Yellow points refer to disorders where the

correlation between region abnormality and SC- and FC-weighted mean neighbour abnormality is significant ($p_{spin} < 0.05$). **c** Left: a region with high abnormality that is also connected to regions with high abnormality is considered a likely disorder epicentre. Middle: epicentre likelihood was calculated as the mean rank of region and neighbour abnormality (see Supplementary Fig. 7 for individual epicentre likelihoods). Right: median epicentre likelihood was calculated for the disorders that show a significant correlation between regional and neighbour abnormality. To limit biasing cross-disorder epicentre likelihood towards epilepsy epicentre likelihood, left and right temporal lobe epilepsy epicentre likelihood was merged into a single mean epicentre likelihood map, prior to calculating the median. For completeness, Supplementary Fig. 8 shows cross-disorder epicentre likelihood when calculated using alternative statistics.

Interestingly, we find that the principal gradient of receptor distribution is particularly dominant towards disease-specific cortical morphology. This receptor gradient represents the maximal variance of density distributions from fourteen receptors and four transporters across nine different neurotransmitter systems, and therefore captures how brain regions may integrate exogenous signals differently[63,70]. This gradient is a powerful predictor of ADHD, left temporal lobe epilepsy, depression, and Parkinson's disease. Indeed, neurotransmitter dysfunction is thought to underlie multiple disorders, including dopamine release in Parkinson's and schizophrenia or serotonin reuptake in depression. Modern therapeutics are designed to selectively manipulate neurotransmitter function for the

purpose of alleviating behavioural symptoms, as opposed to altering brain structure. Our findings confirm the fundamental contribution of neurotransmitters to a wide spectrum of diseases, but they also highlight an important link between the spatial patterning of neurotransmitter receptors and cortical disorder morphology itself[63].

We generally find that cortical abnormality is better predicted by local vulnerability compared to global connectomic vulnerability. One possible reason for the relatively poorer performance of connectivity predictors is that they are generic measures of a region's embedding in a network (number of connections, centrality, connection diversion) but do not consider how this embedding exposes regions to pathology elsewhere in the network. Indeed, we find that disorders whose cortical

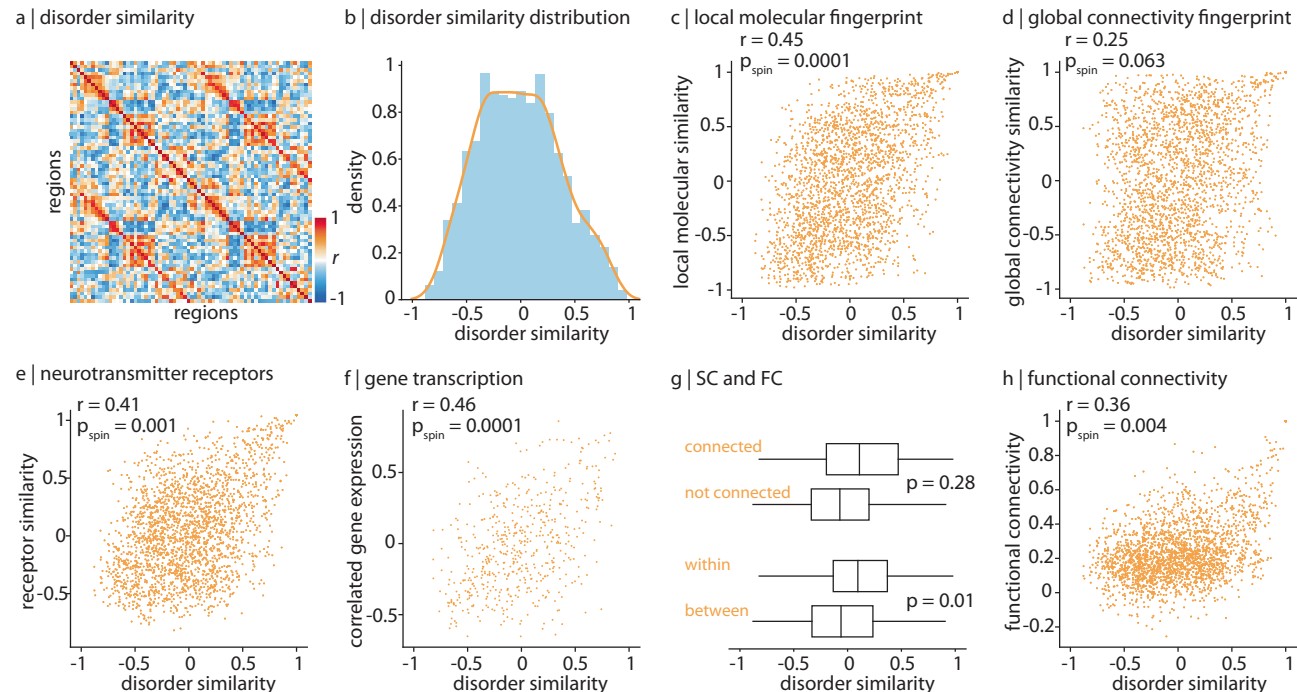

**Fig. 5 | Brain regions with similar molecular annotations are similarly affected across disorders. a** Disorder similarity was computed as the pairwise correlation of regional cortical abnormality across all thirteen disorders such that pairs of regions with high disorder similarity are similarly affected across disorders. **b** A histogram depicting the upper triangle of the disorder similarity matrix. **c** Disorder similarity is significantly correlated to molecular attribute similarity (Pearson's $r(2276) = 0.45$, $p_{spin} = 0.0001$, CI = [0.42, 0.49], two-tailed). **d** Disorder similarity is not significantly correlated with connectomic similarity (Pearson's $r(2276) = 0.25$, $p_{spin} = 0.063$, CI = [0.21, 0.29], two-tailed). **e** Disorder similarity is significantly correlated to neurotransmitter receptor similarity (Pearson's $r(2276) = 0.41$, $p_{spin} = 0.001$, CI = [0.38, 0.45], two-tailed). **f** Left hemisphere disorder similarity is significantly

correlated to correlated gene expression (Pearson's $r(559) = 0.46$, $p_{spin} = 0.0001$, CI = [0.40, 0.53], two-tailed). **g** Disorder similarity is significantly greater within intrinsic functional networks than between networks, against the spin-test ($p = 0.01$; bottom). Disorder similarity is non-significantly greater between structurally connected regions than regions that are not connected, against a degree- and edge-length-preserving null model ($p = 0.028$[101]). Bounds of the box represent the 1st (25%) and 3rd (75%) quartiles, the centre line represents the median, and whiskers represent the minima and maxima of the distribution. $N_{connected} = 592$ edges, $N_{notconnected} = 1686$. $N_{within} = 388$, $N_{between} = 1890$. **h** Disorder similarity is significantly correlated to functional connectivity (Pearson's $r(2276) = 0.36$, $p_{spin} = 0.004$, CI = [0.33, 0.40], two-tailed).

---

morphology is better reflected by local vulnerability also bear a prominent signature of network architecture (e.g., ASD, ADHD, 22q11.2 deletion syndrome, temporal lobe epilepsy, schizotypy, bipolar disorder). This suggests a network-spreading phenomenon where focal pathology or perturbation propagates to connected regions, resulting in cortical abnormality that is correlated with the underlying connection patterns[14]. This interaction between local vulnerability and connectomic vulnerability has previously been reported in neurodegenerative syndromes where the trans-synaptic spreading of misfolded proteins appears to be guided and amplified by local gene expression[10,71–74]. In other words, the poorer performance of connectivity predictors does not suggest that the white-matter architecture is less relevant to disease progression. Indeed, pathogenesis of multiple diseases is thought to originate in the white matter of the brain[14,75,76]. A promising future direction for studying cross-disorder brain abnormalities is to focus on disruptions in white-matter pathways instead of cortical thickness[3,77].

The interaction between molecular vulnerability and network structure naturally raises the question of what are the network epicentres of cortical disorder maps. We find epicentres—regions with high abnormality that are also strongly connected with other regions with high abnormality—in primarily transmodal regions (e.g., inferior temporal lobe, angular gyrus, precuneus, superior parietal cortex), although the motor cortex also appears as an epicentre. That the sensory-motor cortex is an epicentre is consistent with recent reports that multiple psychiatric and neurological disorders are accompanied by sensory deficits and reduced motor control[78–80]. Indeed, the sensory-motor cortex has been previously established as a functional

hub in temporal lobe epilepsies and across multiple psychiatric disorders[68,81]. Interestingly, both the bilateral precuneus and superior parietal cortex are members of the brain's putative rich club—densely inter-connected regions that are thought to support the integration and broadcasting of signals[82]. Rich club regions undergo changes in connectivity patterns in multiple diseases such as schizophrenia, Alzheimer's, and Huntington's[1,3,15,83,84]. We complement this work by showing that the precuneus and superior parietal cortex are both vulnerable to cortical abnormality and, by virtue of their network embedding, increase disease exposure to connected regions. Conversely, although the anterior cingulate cortex (ACC) is implicated across multiple psychiatric disorders[13,85], we do not find that the ACC is an epicentre of cross-disorder cortical morphology. This suggests that although the ACC demonstrates considerable local vulnerability in a subset of brain disorders, it is not consistently involved across the seven disorders included in the epicentre analyses. Altogether, despite variable cortical morphology patterns across the thirteen disorders, when looked at through the lens of network connectivity, we see a more consistent and compact subset of potential epicentres, suggesting greater commonality among diseases than previously appreciated.

One strength of the ENIGMA consortium is that the datasets are pooled over thousands of individuals. However, such large-scale analyses obscure the important inter-subject variability that exists within all disorders. We conduct supplementary analyses in which cortical disorder profiles are stratified by age and disease severity (Supplementary Fig. 4) and find that molecular and connectomic contributions vary. For example, we find that molecular and connectomic

influences on Parkinson's disease differ with disease severity: molecular predictors become less powerful predictors and connectomic predictors become more powerful predictors as the disease progresses. This complements previous work that suggests that atrophy in Parkinson's is the result of network-mediated spread of alpha-synuclein[10]. Furthermore, we find a similar trend of increased connectomic influence for severity of psychotic symptoms: namely, schizotypy and schizophrenia. Although schizotypy is not an earlier stage of schizophrenia (indeed, schizotypy is not a disorder per se but rather a multidimensional continuum of traits related to psychosis), individuals with schizotypy exhibit similar, albeit attenuated, characteristics as schizophrenia patients[35]. We find that the connectomic influence is considerably greater in schizophrenia compared to schizotypy, which may suggest that the structural network gradually becomes more implicated in disease progression. One key factor that we were not able to study in more depth is that of biological sex. Since ENIGMA datasets are all sex-corrected, we are unable to make conclusions about how molecular and connectomic contributions may differ between the sexes. Multiple disorders show well-established sex-differences, including schizophrenia[86], autism[87], and depression[88]. Designing effective clinical interventions will require more nuanced studies that consider the many heterogeneities that exist within each disease.

This work considers multimodal molecular and connectomic contributions to disorders but does not make conclusions about two important features of disease: cognitive phenotypes and genetics. An exciting future direction is to explore whether molecular and connectomic contributions to disease can be related to phenotypic or genetic similarity. Lee et al.[89] compare single-nucleotide polymorphism data across eight psychiatric disorders and find that schizophrenia and bipolar disorder show greatest genetic similarity. This complements our finding that schizophrenia and bipolar disorder have consistent connectomic profiles. Lee et al.[89] also find a clique among the disorders that we find are best predicted by molecular features: ADHD, autism, and major depressive disorder. On the other hand, a comprehensive battery of cognitive and behavioural tests was not uniformly applied to all the disease groups in the ENIGMA datasets. As a result, robust cross-disorder phenotypic profiles are less well-established. Our findings potentially suggest an executive function (anchored by schizophrenia) versus attention (anchored by ADHD and ASD) cognitive axis that separates connectomic versus molecularly informed disorder profiles, but more work is needed to standardise cognitive testing and assess how cognitive/behavioural phenotypes may be related to brain structure. Altogether, future work is necessary to explore how overlapping genetic and neurocognitive disturbances correspond to molecular and connectomic contributions to disease.

The present work should be considered along some important methodological considerations. First, although the ENIGMA consortium standardises pre-processing pipelines and provides large $N$ datasets, allowing for robust results and meaningful comparison between disorder-specific cortical abnormality maps, working with ENIGMA data also has caveats: (1) the measures of cortical abnormality are effect sizes between patients and controls and do not represent tissue volume loss/gain, (2) some of the patient populations included have co-morbidities and patients may be undergoing treatment, including treatment that may have an effect on cortical thickness[90], and (3) all analyses were conducted at the level of 68 cortical brain areas, limiting regional specificity and precluding analyses of the subcortex and cerebellum. Second, despite the fact that structural connectomes were reconstructed from high-resolution diffusion spectrum imaging, diffusion tractography is still prone to false-positives and false-negatives[91–93]. Third, both local molecular and global connectivity predictors are derived from state-of-the-art open-access datasets in healthy participants, but they do not capture individual variability or changes across the lifespan—both of which are key factors in neurological, psychiatric, and neurodevelopmental

disorders. Additionally, the molecular predictors are limited by imaging modality (in particular myelination, for which the T1w/T2w ratio is an indirect estimate[94,95]), and, in the case of the gene and receptor gradients, by the subset of genes and receptors included in the data decomposition. Fourth, we assessed contribution of multiple predictors to disorder maps using simple but robust linear models that are not sensitive to nonlinear contributions or higher-order interactions among the predictors. In addition to this, the correlational structure of the predictor subsets affects predictive power, which limits our ability to compare molecular and connectomic model fits (Supplementary Fig. 15). Fifth, the linear models used in the present analyses assume independence between observations, which is not the case in the brain; we therefore employ spatial-autocorrelation-preserving null models to account for the spatial dependencies between regions throughout the report. Finally, although the present report spans a wide range of neurological, psychiatric, and neurodevelopmental disorders, results are only valid for this subset of disorders. Future work is needed to map local and global vulnerabilities to the many more brain diseases and disorders that exist.

In summary, we find that molecular and connectomic vulnerability jointly shape cross-disorder cortical abnormalities. Cross-disorder regional vulnerability is largely driven by molecular fingerprints, including neurotransmitter receptor densities and gene expression, while connection patterns among vulnerable regions further amplify this vulnerability. Our results highlight how an integrative, multimodal approach can illuminate the contributions of local biology and connectome architecture to brain disease.

## Methods

### Cortical disorder maps
Patterns of cortical thickness were collected for the available thirteen neurological, neurodevelopmental, and psychiatric disorders from the ENIGMA consortium and the *enigma* toolbox (https://github.com/MICA-MNI/ENIGMA[96]), including: 22q11.2 deletion syndrome ($N = 474$ participants, $N = 315$ controls)[26], attention-deficit/hyperactivity disorder (ADHD; $N = 733$ participants, $N = 539$ controls)[27], autism spectrum disorder (ASD; $N = 1571$ participants, $N = 1651$ controls)[28], idiopathic generalised ($N = 367$ participants), right temporal lobe ($N = 339$ participants), and left temporal lobe ($N = 415$ participants) epilepsies ($N = 1727$ controls)[29], depression ($N = 2148$ participants, $N = 7957$ controls)[30], obsessive-compulsive disorder (OCD; $N = 1905$ participants, $N = 1760$ controls)[31], schizophrenia ($N = 4474$ participants, $N = 5098$ controls)[32], bipolar disorder ($N = 1837$ participants, $N = 2582$ controls)[33], obesity ($N = 1223$ participants, $N = 2917$ controls)[34], schizotypy ($N = 3004$ participants)[35], and Parkinson's disease ($N = 2367$ participants, $N = 1183$ controls)[36]. The ENIGMA (Enhancing Neuroimaging Genetics through Meta-Analysis) Consortium is a data-sharing initiative that relies on standardised processing and analysis pipelines, such that disorder maps are comparable[25]. Altogether, over 21,000 participants were scanned across the thirteen disorders, against almost 26,000 controls. The analysis was limited to adults in all cases except ASD where the cortical abnormality map is only available aggregated across all ages (2–64 years). The values for each map are $z$-scored effect sizes (Cohen's $d$) of cortical thickness in patient populations versus healthy controls. Imaging and processing protocols can be found at http://enigma.ini.usc.edu/protocols/, and detailed demographic information can be found in the supplement of each accompanying article. Local review boards and ethics committees approved each individual study separately, and written informed consent was provided according to local requirements.

### Structural and functional networks
**Lausanne dataset.** Structural and functional data were collected at the Department of Radiology, University Hospital Center and University of Lausanne, on $n = 70$ healthy young adults (16 females, $25.3 \pm 4.9$

years)[97]. Informed consent was obtained from all participants and the protocol was approved by the Ethics Committee of Clinical Research of the Faculty of Biology and Medicine, University of Lausanne. The scans were performed in a 3-T MRI scanner (Trio; Siemens Medical), using a 32-channel head coil. The protocol included (1) a magnetisation-prepared rapid acquisition gradient echo (MPRAGE) sequence sensitive to white/grey matter contrast (1 mm in-plane resplution, 1.2 mm slice thickness), (2) a DSI sequence (128 diffusion-weighted volumes and a single b0 volume, maximum $b$-value 8000 s/mm$^2$, $2.2 \times 2.2 \times 3.0$ mm voxel size), and (3) a gradient echo-planar imaging (EPI) sequence sensitive to blood-oxygen-level-dependent (BOLD) contrast (3.3 mm in-plane resolution and slice thickness with a 0.3 mm gap, TR 1920 ms, resulting in 280 images per participant). Participants were not subject to any overt task demands during the fMRI scan. The Lausanne dataset is available at https://zenodo.org/record/2872624#.XOJqE99fhmM and has been used in other work[98,99].

Grey matter was parcelled according to the 68-region Desikan-Killiany cortical atlas[40]. Structural connectivity was estimated for individual participants using deterministic streamline tractography. The procedure was implemented in the Connectome Mapping Toolkit[100], initiating 32 streamline propagations per diffusion direction for each which matter voxel. Collating each individual's structural connectome was done using a group-consensus approach that seeks to preserve the density and edge-length distributions of the individual connectomes (see *Group-consensus structural network*[101]). The binary density for the final whole-brain structural connectome was 24.6%. For the weighted structural connectome, edges were weighted by the average log-transform of non-zero streamline density, scaled to values between 0 and 1.

Functional MRI data were pre-processed using procedures designed to facilitate subsequent network exploration[102]. fMRI volumes were corrected for physiological variables, including regression of white matter, cerebrospinal fluid, and motion (3 translations and 3 rotations, estimated by rigid body coregistration). BOLD time-series were then subjected to a low-pass filter (temporal Gaussian filter with full width at half maximum equal to 1.92 s). The first four time points were excluded from subsequent analysis to allow the time-series to stabilise. Motion scrubbing was performed as described by ref. 102. The data were parcelled according to the same 68-region Desikan-Killiany atlas used for the structural network. Individual functional connectivity matrices were defined as zero-lag Pearson correlation among the fMRI BOLD time-series. A group-consensus functional connectivity matrix was estimated as the mean connectivity of pairwise connections across individuals. Note that one individual did not undergo an fMRI scan and therefore the functional connectome was composed of $n = 69$ participants.

**Human Connectome Project.** Following the procedure described in de Wael et al.[103], we obtained structural and functional magnetic resonance imaging (MRI) data for 326 unrelated participants (age range 22–35 years, 145 males) from the Human Connectome Project (HCP; S900 release[104]; informed consent obtained). All four resting-state fMRI scans (two scans (R/L and L/R phase encoding directions) on day 1 and two scans (R/L and L/R phase encoding directions) on day 2, each about 15 min long; TR = 720 ms), as well as diffusion-weighted imaging (DWI) data were available for all participants. All the structural and functional MRI data were pre-processed using HCP minimal pre-processing pipelines[104,105]. We provide a brief description of data pre-processing below, while detailed information regarding data acquisition and pre-processing is available elsewhere[104,105].

DWI data was pre-processed using the MRtrix3 package[106] (https://www.mrtrix.org/). More specifically, fibre orientation distributions were generated using the multi-shell multi-tissue constrained spherical deconvolution algorithm from MRtrix[107,108]. White-

matter edges were then reconstructed using probabilistic streamline tractography based on the generated fibre orientation distributions[109]. The tract weights were then optimised by estimating an appropriate cross-section multiplier for each streamline following the procedure proposed by Smith et al.[110] and a connectivity matrix was built for each participant using the 68-region Deskian-Killiany parcellation[40,41]. Collating each individual's structural connectome was done using a group-consensus approach that seeks to preserve the density and edge-length distributions of the individual connectomes (see *Group-consensus structural network*[101]). The binary density for the final whole-brain structural connectome was 31.2%. For the weighted structural connectome, edges were weighted by the average log-transform of non-zero streamline density, scaled to values between 0 and 1.

All 3T functional MRI time-series were corrected for gradient nonlinearity, head motion using a rigid body transformation, and geometric distortions using scan pairs with opposite phase encoding directions (R/L, L/R)[103]. Further pre-processing steps include coregistration of the corrected images to the T1w structural MR images, brain extraction, normalisation of whole-brain intensity, high-pass filtering (>2000s FWHM; to correct for scanner drifts), and removing additional noise using the ICA-FIX process[103,111]. The pre-processed time-series were then parcelled to 68 cortical brain regions according to the Desikan-Killinay atlas[40,41]. The parcelled time-series were used to construct functional connectivity matrices as a Pearson correlation coefficient between pairs of regional time-series for each of the four scans of each participant. A group-average functional connectivity matrix was constructed as the mean functional connectivity across all individuals and scans.

## Group-consensus structural network

To construct a representative group-level connectome, we used a consensus approach that seeks to preserve the density and edge-length distributions of the individual connectomes (first applied in Mišić et al.[112] and presented formally in Betzel et al.[101]). This procedure better captures important organisational features of subject-level networks compared to other consensus methods (i.e., thresholding based on whether an edge is observed in a fraction of subjects)[101]. The procedure for generating the consensus network is as follows. First, existing edges across participants were binned according to length. The number of bins was determined heuristically as the square root of the mean binary density across participants. Within each bin, the $k$ most frequently occurring edges across participants were retained. $k$ was set as the average across the number of edges each participant has in the bin. To ensure that interhemispheric edges are not under-represented, we carried out this procedure separately for inter- and intrahemispheric edges.

## Molecular predictors

A total of seven local molecular predictors were used in the multilinear model to represent the influence that local molecular attributes have on disorder-specific cortical morphology.

**Gene expression gradient.** The first principal component of gene expression ("gene gradient") was used to represent the variation in gene expression levels across the left cortex. This gradient has been previously related to cell type distributions and cell-specific gene expression, which suggests the gradient is related to the cellular architecture of the brain[16,20,37–39]. Gene expression data was collected by the Allen Human Brain Atlas as described in Hawrylycz et al.[37] and processed by *abagen*, an open-source Python toolbox[113]. A total of 11,560 genes with differential stability greater than 0.1 were retained in the region by gene matrix[114]. The left gene gradient was mirrored in the right hemisphere. A detailed account of the specific processing choices made can be found in Hansen et al.[39].

**Receptor gradient.** The first principal component of receptor density ("receptor gradient") was used to represent the variation in receptor densities across the cortex. Receptor densities were estimated using PET tracer studies for a total of 18 receptors and transporters, across 9 neurotransmitter systems. These include dopamine ($D_1$[115], $D_2$[116–119], DAT[120]), norepinephrine (NET[121–124]), serotonin (5-HT$_{1A}$[125], 5-HT$_{1B}$[125–132], 5-HT$_{2A}$[133], 5-HT$_4$[133], 5-HT$_6$[134,135], 5-HTT[133]), acetylcholine ($\alpha_4\beta_2$[132,136], $M_1$[137], VAChT[138,139]), glutamate (mGluR$_5$[140,141]), GABA (GABA$_A$[142]), histamine ($H_3$[143]), cannabinoid (CB$_1$[144–147]), and opioid (MOR[148]). Volumetric PET images were registered to the MNI-ICBM 152 nonlinear 2009 (version c, asymmetric) template, averaged across participants within each study, then parcellated to 68 cortical regions. Parcellated PET maps were then $z$-scored before compiling all receptors/transporters into a region × receptor matrix of relative densities. Data were originally presented as an atlas in Hansen et al.[63].

**Excitatory-inhibitory ratio.** The excitatory-inhibitory ratio was computed as the ratio of $z$-scored PET-derived excitatory to inhibitory neurotransmitter receptor densities in the cortex, using the same dataset that was used to compute the receptor gradient. Excitatory neurotransmitter receptors included are: 5-HT2$_A$, 5-HT$_4$, 5-HT$_6$, $D_1$, mGluR$_5$, $\alpha_4\beta_2$, and $M_1$. Inhibitory neurotransmitter receptors included are: 5-HT$_{1A}$, 5-HT$_{1B}$, CB$_1$, $D_2$, GABA$_A$, $H_3$, and MOR.

**Glycolytic index.** Aerobic glycolysis is the process of converting glucose to lactate in the presence of oxygen. It is traditionally calculated as the ratio of oxygen metabolism to glucose metabolism. Here, we use glycolytic index, a measure of aerobic glycolysis that mitigates certain limitations of using the traditional ratio[149]. Glycolytic index is defined as the residual after fitting glucose metabolism to oxygen metabolism in a linear regression model. Larger values indicate more aerobic glycolysis. Note that glycolytic index and the traditional ratio are highly correlated (see Vaishnavi et al.[149]). Data were collected, calculated, and made available by Vaishnavi et al.[149]. Glucose metabolism was obtained as described in the section below, and oxygen metabolism was collected in the same participants by administering [$^{15}$O]-labelled water, carbon monoxide, and oxygen. All experiments were approved by the Human Research Protection Office and the Radioactive Drug Research Committee at Washington University in St. Louis. Written informed consent was provided by all participants.

**Glucose metabolism.** Glucose metabolism in the cortex was measured in 33 healthy adults (19 female, mean age $25.4 \pm 2.6$ years) by administering [$^{18}$F]-labelled fluorodeoxyglucose (FDG) for a PET scan, as described in detail in Vaishnavi et al.[149]. All experiments were approved by the Human Research Protection Office and the Radioactive Drug Research Committee at Washington University in St. Louis. Written informed consent was provided by all participants.

**Synapse density.** Synapse density in the cortex was measured in 76 healthy adults (45 males, $48.9 \pm 18.4$ years of age) by administering [$^{11}$C]UCB-J, a PET tracer that binds to the synaptic vesicle glycoprotein 2A (SV2A)[150–161]. Data were collected on an HRRT PET camera for 90 min post injection. Non-displaceable binding potential (BP$_{ND}$) was modelled using SRTM2, with the centrum semiovale as reference and $k'$ fixed to 0.027 (population value). Each study was performed under a protocol approved by the Yale University Human Investigation Committee and the Yale New Haven Hospital Radiation Safety Committee, and written informed consent was obtained from all participants.

**Myelination.** Data from the Human Connectome Project (HCP, S1200 release)[104,105] was used for measures of T1w/T2w ratios—a proxy for intracortical myelin—for 417 unrelated participants (age range 22–37 years, 193 males), as approved by the WU-Minn HCP Consortium. Images were acquired on a Siemens Skyra 3T scanner, and included a T1-weighted MPRAGE sequence at an isotropic resolution of 0.7 mm, and a T2-weighted SPACE also at an isotropic resolution of 0.7 mm. Details on imaging protocols and procedures are available at http://protocols.humanconnectome.org/HCP/3T/imaging-protocols.html. Image processing includes correcting for gradient distortion caused by non-linearities, correcting for bias field distortions, and registering the images to a standard reference space. T1w/T2w ratios for each participant was made available in the surface-based CIFTI file format and parcellated into 68 cortical regions according to the Lausanne anatomical atlas[41]. Note that the T1w/T2w ratio is an MRI-based estimate of myelin content that has not yet been validated against myelin histology[94]. Other MRI-based proxies may be more suitable alternatives, such as magnetisation transfer or simultaneous tissue relaxometry of R1 and R2 relaxation rates and proton density (SyMRI), which have been validated using myelin histology and are closely correlated to one another[94,95]. Additionally, PET imaging may be a promising avenue for mapping myelin content in the brain[162,163].

## Connectivity predictors

A total of nine global connectome predictors were used in the multilinear model to represent the influence that global connectivity has on disorder-specific cortical morphology. In the main text, connectome measures were computed on the weighted structural connectome. Analyses were repeated using a binary structural connectome and an absolute functional connectome (Supplementary Fig. 12). All connectivity measures were computed using the Python-equivalent of the Brain Connectivity Toolbox, *bctpy*.

**Strength.** The strength of region $i$ is the sum of the edges connected to region $i$. For a binary structural connectome, the strength is equivalent to the degree, which is the number of links connected to region $i$.

**Betweenness centrality.** Betweenness centrality of region $i$ is the fraction of all shortest paths between any two regions that traverse region $i$.

**Closeness centrality.** Closeness centrality is equivalent to the mean shortest path distance from region $i$ to every other region in the network.

**Euclidean distance.** Mean Euclidean distance of a region to all other regions in the network represents how spatially close one region is to all other regions.

**Participation coefficient.** Participation coefficient was computed using the putative intrinsic functional networks of the brain[164]. Participation coefficient represents the connection diversity of a region. A region with high participation coefficient is well connected to several different networks, whereas a region with low participation coefficient primarily makes local (within-network) connections.

**Clustering coefficient.** The clustering coefficient of region $i$ is the fraction of all triangles that are around region $i$. Equivalently, it is the fraction of all of region $i$'s neighbours that are also neighbours with each other. In the case of the weighted structural connectome, clustering coefficient is the average geometric mean of all triangles associated with the region.

**Mean first passage time.** The mean first passage time from region $i$ to $j$ is the expected amount of time it takes a random walker to reach region $j$ from $i$ for the first time. For each region, mean first passage time was averaged across regions, resulting in a mean mean first passage time representing the average amount of time it takes a random walker to travel from region $i$ to any other region in the network for the first time.

## Temporal predictors

Six-minute resting-state eyes-open magenetoencephalography (MEG) time-series were acquired from the Human Connectome Project (HCP, S1200 release) for 33 unrelated subjects (age range 22–35, 17 males)[104,105]. Complete MEG acquisition protocols can be found in the HCP S1200 Release Manual. For each subject, we computed the power of the run at the vertex level across six different frequency bands: delta (2–4 Hz), theta (5–7 Hz), alpha (8–12 Hz), beta (15–29 Hz), low gamma (30–59 Hz), and high gamma (60–90 Hz), using the open-source software, Brainstorm[165]. Each power band was then parcellated into 68 cortical regions[41].

## Dominance analysis

Dominance analysis seeks to determine the relative contribution ("dominance") of each input variable to the overall fit (adjusted $R^2$) of the multiple linear regression model (https://github.com/dominance-analysis/dominance-analysis[42,43]). This is done by fitting the same regression model on every combination of input variables ($2^p - 1$ submodels for a model with $p$ input variables). Total dominance is defined as the average of the relative increase in $R^2$ when adding a single input variable of interest to a submodel, across all $2^p - 1$ submodels. The sum of the dominance of all input variables is equal to the total adjusted $R^2$ of the complete model, making total dominance an intuitive measure of contribution. Note that significance testing is not applied to the individual dominances because the contributions of input variables are relative to other predictors in the model and input variables do not act in isolation.

Each multilinear model was cross-validated using a distance-dependent method proposed by Hansen et al.[39]. Briefly, for each of 1000 iterations, the 75% of regions closest in Euclidean distance to a randomly chosen source node were selected as the training set, and the remaining 25% of regions as the test set. Predicted values in the test set were then correlated to true abnormality patterns, and the correlations are shown in Supplementary Fig. 2.

## Network spreading

Network spreading was computed as first introduced in Shafiei et al.[13] and later adopted in Chopra et al.[54], Shafiei et al.[9]. Briefly, regional abnormality was defined as the normalised effect size used in all ENIGMA brain maps. For each region $i$, its neighbours are those with which region $i$ is connected via a structural connection, as defined by the structural connectivity matrix. Mean neighbour abnormality of region $i$ ($D_i$) is the average abnormality of region $i$'s neighbours, where $d_j$ represents the abnormality of neighbour $j$. Notably, this method normalises neighbour abnormality by the number of connections made by region $i$ ($N_i$).

$$D_i = \frac{1}{N_i} \sum_{j \neq i, j=1}^{N_i} d_j \times SC_{ij} \tag{1}$$

when neighbour abnormality is weighted by functional connectivity, each neighbour's abnormality are weighted by the functional connection to node $i$ ($FC_{ij}$).

$$D_i = \frac{1}{N_i} \sum_{j \neq i, j=1}^{N_i} d_j \times SC_{ij} \times FC_{ij} \tag{2}$$

Each brain region was assigned a rank in terms of their node abnormality and their mean neighbour abnormality. The average of node and neighbour abnormality ranks was defined as the epicentre likelihood of the node, where nodes with high abnormality and whose neighbours are also highly atypical are more likely to be an epicentre of the disorder.

## Disorder similarity

For every brain region, we constructed a 13-element vector of disorder abnormality, where each element represents a disorder's cortical abnormality at the region. For every pair of brain regions, we correlated the abnormality vectors to quantify how similarly two brain regions are affected across disorders. This results in a region-by-region matrix of "disorder similarity" (Fig. 5a). We verified that no single disorder pattern was driving the disorder similarity matrix by recalculating the disorder similarity when a single disorder is excluded. We then correlated the leave-one-out disorder similarity matrix with the original disorder similarity matrix. The minimum correlation was $r = 0.95$ (Supplementary Fig. 16a). Finally, influence on the disorder similarity matrix by a disorder $i$ was quantified as

$$I_i = 1 - \text{corr}(D, D_i) \tag{3}$$

where $D$ is the original disorder similarity matrix and $D_i$ is the disorder similarity matrix constructed when disorder $i$ is excluded (Supplementary Fig. 16b).

## Null models

Spatial-autocorrelation-preserving permutation tests were used to assess statistical significance of associations across brain regions, termed "spin tests"[166–168]. We created a surface-based representation of the parcellation on the FreeSurfer fsaverage surface, via files from the Connectome Mapper toolkit (https://github.com/LTS5/cmp). We used the spherical projection of the fsaverage surface to define spatial coordinates for each parcel by selecting the coordinates of the vertex closest to the center of the mass of each parcel[98]. These parcel coordinates were then randomly rotated, and original parcels were reassigned the value of the closest rotated parcel (1000 repetitions). Parcels for which the medial wall was closest were assigned the value of the next most proximal parcel instead. The procedure was performed at the parcel resolution rather than the vertex resolution to avoid upsampling the data, and to each hemisphere separately. This spin-permuted null model involves conflating and collapsing the brain surface to and from a sphere. The geometry of the cortical surface is therefore not retained in the spinning process, which may result in null distributions that are too wide. Other methods for constructing spatial null models exist, such as generative models[169] and 2D spatial wavestrapping[170].

A second null model was used to test whether disorder similarity is greater in connected regions than unconnected regions. This model generates a null structural connectome that preserves the density, edge length, and degree distributions of the empirical structural connectome[66,171,172]. Briefly, edges were binned according to Euclidean distance. Within each bin, pairs of edges were selected at random and swapped. This procedure was then repeated 10,000 times. To compute a $p$-value, the mean disorder similarity of unconnected edges was subtracted from the mean disorder similarity of connected edges, and this difference was compared to a null distribution of differences computed on the rewired networks.

## Reporting summary

Further information on research design is available in the Nature Research Reporting Summary linked to this article.

# Data availability

All data used in the present report is openly available at https://github.com/netneurolab/hansen_crossdisorder_vulnerability. More specifically, ENIGMA datasets are available through the ENIGMA consortium and the ENIGMA toolbox (https://github.com/MICA-MNI/ENIGMA[96]). The Lausanne dataset is available at https://zenodo.org/record/2872624#.XOJqE99fhmM[97]. The HCP dataset is available at https://db.humanconnectome.org/. Molecular predictors are available as

volumetric images in the `neuromaps` toolbox (https://netneurolab.github.io/neuromaps/[173]). The Allen Human Brain Atlas is available at https://human.brain-map.org/.

## Code availability

All code used to perform the analyses can be found at https://github.com/netneurolab/hansen_crossdisorder_vulnerability and on Zenodo (https://doi.org/10.5281/zenodo.6795748).

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

## Acknowledgements

This research was undertaken thanks in part to funding from the Canada First Research Excellence Fund, awarded to McGill University for the Healthy Brains for Healthy Lives initiative. B.M. acknowledges support from the Natural Sciences and Engineering Research Council of Canada (NSERC Discovery Grant RGPIN #017-04265), the Canada Research Chairs Programme, the Brain Canada Future Leaders Fund and the Healthy Brains for Healthy Lives initiative. J.Y.H. acknowledges support from the Helmholtz International BigBrain Analytics & Learning Laboratory, the Natural Sciences and Engineering Research Council of Canada, and the Fonds de reserches de Québec. The research studies produced by the ENIGMA Working Groups would not be possible without the contributions of many researchers across the globe and the authors of this work thank all scientists who contribute to making this work possible. A full list of ENIGMA Consortium current and past members can be found here http://enigma.ini.usc.edu/ongoing/members/. The authors acknowledge the NIH Big Data to Knowledge (BD2K) award for foundational support and consortium development (U54 EB020403 to P.M.T.) and support from NIMH R01MH116147 (P.M.T.), NIMH R01MH116147 (T.G.M.v.E.), NIMH R01 MH117601 (N.J., L.S.), NIMH R01MH085953 (C.E.B.), NIMH R21MH116473 (C.E.B.), NIMH 1U01MH119736 (C.E.B.). For a complete list of ENIGMA-related grant support please see here: http://enigma.ini.usc.edu/about-2/funding. J.B. has been supported by the EU-AIMS (European Autism Interventions) and AIMS-2-TRIALS programmes, which receive support from Innovative Medicines Initiative Joint Undertaking Grant No. 115300 and 777394, the resources of which are composed of financial contributions from the European Union's FP7 and Horizon2020 Programmes, and from the European Federation of Pharmaceutical Industries and Associations (EFPIA) companies' in-kind contributions, and AUTISM SPEAKS, Autistica and SFARI; and by the Horizon2020 supported programme CANDY Grant No. 847818). B.F. is supported by the European Community's Horizon 2020 Programme (H2020/2014-2020) under grant agreements no. 667302 (CoCA), no. 728018 (Eat2beNICE), and no. 847879 (PRIME). This work was supported by a personal Veni grant to M.H. from the Netherlands Organisation for Scientific Research (NWO, grant number 91619115). C.R.M. is supported by NIH R01 NS065838; R21 NS107739. D.J.S. is supported by South African Medical Research Council. G.M. is funded by a Wellcome Trust & The Royal Society Sir Henry Dale Fellowship [202397/Z/16/Z]. The funders had no role in study design, data collection and analysis, decision to publish or preparation of the manuscript.

## Author contributions

J.Y.H. and B.M. conceived the study and wrote the manuscript, with valuable revision from all authors. J.Y.H. performed the formal analysis, with contribution from G.S. J.Y.H. interpreted the results with contribution from G.S., J.W.V., A.D., and B.M. K.S., C.E.B., M.H., B.F., D.V., J.B., C.R.M., S.M.S., L.S., D.J.V., O.A.V., D.J.S., T.G.M.v.E, C.R.K.C., O.A.A., T.H., N.O., G.M., A.A., Y.V., N.J., S.I.T., P.M.T., and R.E.C. provided data. B.M. was the project administrator.

## Competing interests

C.R.K.C., N.J., P.M.T. received partial research support from Biogen, Inc., for research unrelated to this manuscript. J.B. has been in the past 3 years a consultant to/member of advisory board of/and/or speaker for Takeda/Shire, Roche, Medice, Angelini, Janssen, and Servier. He is not an employee of any of these companies, and not a stock shareholder of any of these companies. He has no other financial or material support, including expert testimony, patents, royalties. B.F. has received educational speaking fees from Medice GmbH. D.J.S. has received research grants and/or consultancy honoraria from Lundbeck and Sun. The remaining authors declare no competing interests.

## Additional information

[1]McConnell Brain Imaging Centre, Montréal Neurological Institute, McGill University, Montréal, QC, Canada. [2]Department of Psychiatry, Perelman School of Medicine, University of Pennsylvania, Philadelphia, PA, USA. [3]Department of Radiology and Biomedical Imaging, Yale School of Medicine, New Haven, CT 06520, USA. [4]Departments of Psychiatry and Biobehavioral Sciences and Psychology, Semel Institute for Neuroscience and Human Behavior, University of California, Los Angeles, CA, USA. [5]Departments of Psychiatry and Human Genetics, Radboud University Medical Center, Nijmegen, The Netherlands.

[6]Donders Institute for Brain, Cognition and Behavior, Radboud University, Nijmegen, The Netherlands. [7]Department of Psychiatry, University of California San Diego, La Jolla, CA, USA. [8]Department of Clinical and Experimental Epilepsy, UCL Queen Square Institute of Neurology, London WC1N 3BG, UK. [9]Centre for Youth Mental Health, The University of Melbourne, Melbourne, VIC, Australia. [10]Department of Psychiatry, Amsterdam UMC, Vrije Universiteit Amsterdam, Amsterdam Neuroscience, Amsterdam, The Netherlands. [11]Department of Anatomy & Neuroscience, Amsterdam UMC, Vrije Universiteit Amsterdam, Amsterdam Neuroscience, Amsterdam, the Netherlands. [12]SA MRC Unit on Risk & Resilience in Mental Disorders, Dept of Psychiatry & Neuroscience Institute, University of Cape Town, Cape Town, South Africa. [13]Clinical Translational Neuroscience Laboratory, Department of Psychiatry and Human Behavior, & Center for the Neurobiology of Leaning and Memory, University of California Irvine, 309 Qureshey Research Lab, Irvine, CA, USA. [14]Keck School of Medicine, Imaging Genetics Center, Mark and Mary Stevens Neuroimaging and Informatics Institute, University of Southern California, Los Angeles, CA, USA. [15]NORMENT Centre, Institute of Clinical Medicine, University of Oslo and Division of Mental Health and Addiction, Oslo University Hospital, Oslo, Norway. [16]Department of Psychiatry, Dalhousie University, Halifax, NS, Canada. [17]Institute of Translational Psychiatry, University of Münster, Münster, Germany & Department of Psychiatry, Jena University Hospital/Friedrich-Schiller-University Jena, Jena, Germany. [18]Department of Psychosis Studies & MRC Centre for Neurodevelopmental Disorders, King's College London, London, UK. [19]Department of Biomedical Sciences of Cells and Systems, University of Groningen, Groningen, The Netherlands. ✉e-mail: bratislav.misic@mcgill.ca

