## [Peer Review File · Nature Communications]

Local molecular and global connectomic contributions to cross-disorder cortical abnormalitiesREVIEWER COMMENTS

Reviewer #1 (Remarks to the Author):

“Molecular and connectomic vulnerability shape cross-disorder cortical abnormalities” by Hansen et al uses ENIGMA data to build a library of cortical thickness anomalies across 13 neurological and psychiatric disorders and compare these to molecular and connectivity fingerprints. The authors first ask whether each disorder, considered individually, aligns more closely to molecular gradients or regional connection topologies. They next build a transdiagnostic composite map and ask whether this has a stronger molecular or connectivity association.

The paper provides a novel approach to understanding transdiagnostic neurobiological perturbations to cortical grey matter, and their relationship to genetic and connectomic features. It provides a platform for future work to integrate other disorders or address developmental or ageing processes.

The paper is interesting, address a question of fundamental and clinical interest using advanced computational techniques. I think it would be of substantial interest to the readers of Nature Communications. I have mainly observations for consideration which the authors may choose to address with additional work, or through a deeper discussion of current limitations.

1. Each of the disorders is associated with a phenotypic profile (symptoms + neurocognitive disturbances). The study is somewhat limited for not incorporating these. For example, are there common/overlapping neurocognitive disturbances that correspond to the overlapping neurobiological changes. I assume ENIGMA has this information per disorder, that could be incorporated alongside the cortical thickness. This would extend the paper more deeply into translational territory and would be stronger than the superficial “reverse inference” observations on the top of page 8, second column. Likewise, are there symptom- or cognitive features that associate more strongly with the “molecular-correlated” disorders on Fig. 3 (ADHD, MDD – e.g. attention) versus those with a “connectivity” profile (SCZ – e.g. executive functioning). If this is to be deferred to future work, it should in the least be addressed in the Discussion.

2. Most of the disorders – some more than others – are quite heterogenous over time (recent onset versus chronic), across age, and across patients (e.g. unipolar versus melancholic versus bipolar depression). Most of the patients in the clinical ENIGMA cohorts are quite stable and have been on pharmacological therapies, many of which have impacts on cortical thickness (e.g. anti-psychotics [1]). MDD, SCZ and ADHD in particular are not unitary entities across time or patients and their representation as singular entities in the various figures is a translational limitation. This approach is not at all unique to the present paper, but a widespread limitation so there is no line to be drawn in the sand here: Again, this could be addressed with some additional analyses (such as a break-out analysis of medication, age or time since illness onset analyses on at least one diagnostic category) or simply acknowledged in the Discussion

3. A methodological limitation is that the core method essentially derives from “correlations of correlations”, limiting insights into deeper mechanistic causes. I think the existing “dig down” analyses are elucidating and largely overcome this weakness, but I would avoid statements like “regional molecular vulnerability and macroscale brain network architecture interact to drive the spatial patterning of cortical abnormalities in multiple disorders” which is something that could hardly have a counter-factual.

4. The main marker of neurobiological perturbation is cortical thickness: While this is a good choice, it is arguably not that surprising to see molecular associations stronger than connectomic ones, given that the former impact directly on cortical activity, function and integrity. But does intra-cortical pathology hold a privileged position when considering the pathogenesis of brain disorders? Some might argue that many disorders arise primarily in white matter connections, and that these then

propagate to cortical grey matter. This is a relatively minor point but might warrant a brief note in the paper.

Minor:

4. I thought it was only the analysis pipelines that were harmonized across ENIGMA not the image acquisition (p9) which were quite site specific (but I could be wrong!).

5. The overall (anti-)correlations among the graph metrics are stronger than the molecular features (heatmap of Figure 1), which likely has an impact on how the relative and adjusted R^2 values are derived using the dominance analysis (as the effective degrees of freedom differ and there does not seem to be a step to partial out mutual correlations).

6. p4: Why “embryonic” stage. Wiring abnormalities can arise across any neurodevelopmental stage.

7. The MRI data used to reconstruct the normative functional and structural connectivity data is not particularly state of the art (long TR, 32 channel head coil, single-shell, deterministic tractography etc). I doubt this will impact on the bottom-line findings, but I am not sure why the authors did not choose a more advanced HCP-like data set to work with.

8. The authors note the limitations of the 68-region DK cortical atlas – and the rationale for using it. But the regions in this atlas differ highly in extent and it would be worth checking that regions that dominate the cross-diagnostic findings are not simply larger in extent and therefore more likely to overlap by chance, and more likely to have higher SnR for all the analysis steps.

9. A point perhaps for another occasion, but I am a bit sceptical about the now widely used spin test to represent nulls for spatial correlations between data sets: While it is crucial that spatial correlations be preserved in nulls, the process of conflated and then collapsing the cortical surface will introduce geometric distortions that over-estimate the true underlying null variance: spatial correlations from highly curved regions will be compressed onto the inflated representation which are then spun and mapped onto flatter regions. While it seems likely that spatial correlations will be preserved ‘on average’ the variance (the width of the null distribution) will be a composite of the true variance plus these geometric effects. A more accurate null could be obtained through whitening then resampling on the naïve cortical geometry. I do not think this is an issue for the present paper (because if anything a wider null means a more conservative test), but hope this limitation either be rebutted or acknowledged in the corresponding Methods section of the ms (p12).

References:

1. Voineskos, A. N., Mulsant, B. H., Dickie, E. W., Neufeld, N. H., Rothschild, A. J., Whyte, E. M., ... & Flint, A. J. (2020). Effects of antipsychotic medication on brain structure in patients with major depressive disorder and psychotic features: neuroimaging findings in the context of a randomized placebo-controlled clinical trial. *JAMA psychiatry*, 77(7), 674-683.

Reviewer #2 (Remarks to the Author):

This manuscript is a rigorous and methodologically sound study that addresses a key neuroscience question. The authors present a relevant set of results that will be of major interest to the research community. Evidently, one of the strongest aspects of the study is the data that has been exploited. Although this (ENIGMA) data has been extensively analysed before in previous studies (cited by the authors), the exhaustive coverage of a wide range of local (molecular) and global (connectivity) metrics using other public datasets demonstrates how integrative approaches can provide new insights on the molecular underpinnings of structural brain alterations associated to brain disease. First of all, I want to congratulate the authors for the important and valuable investment that they have

made with the Code and Data repository. The null models and statistical procedures are appropriate (spin test, cross-validations, etc.). Guidelines are clear, the code is commented and there is an honest effort to facilitate the replicability of their findings. Despite the large number of results, they are all clearly explained and detailed. The schematics are very illustrative and the narrative is easy to follow. I have some comments that I hope can contribute to improving the manuscript:

For some analyses, a considerable number of tests have been carried out (for example those depicted in Figure 2). I understand that correcting for multiple comparisons could be problematic here (metrics are not independent) but, if no correction procedure is used I think it should be at least acknowledged on the limitation section.

Is the epicentre likelihood related to Euclidean distance? I found intriguing that the highest epicentre likelihood is located at motor (and parietal) cortex. I would have expected that association regions and hubs will constitute these epicentres. As authors point out “In particular, regions that are highly connected and potentially important for communication tend to be disproportionately affected by disease [26, 114].” I wonder whether I am misinterpreting the metric or whether there the distance effect on the connectivity matrix influences the epicentre likelihood. Interestingly, authors discuss this finding in the discussion but arguments are focused on previous evidence showing that sensory-motor cortex is a plausible epicentre. Could you please discuss whether this is specific to primary cortex? or rather there is also evidence of the opposite (epicentres predominantly in associations cortices). Note: Looking at Figure S7, I think that one possibility is that this is driven by the epilepsy (which is duplicated). Given that these maps show considerable differences in regional distribution across conditions, is the average a meaningful metric?

The disorder similarity matrix showed in Figure 5a looks very similar to a Structural Covariance Network. Given that it is computed as “pairwise correlation of regional cortical thickness” (according to figure 5a legend) I wonder whether the associations described later are structural-Covariance-related (e.g. as described in Romero-Garcia et al. 2018, Neuroimage) rather than disorder-related. Would it make sense to compute the disorder similarity as the correlation between “cortical thinning (for example, the difference with controls or the T-scores)” rather than using raw cortical thickness? Note: The Disorder similarity methods section states that “we correlated the abnormality vectors” so authors may already have gone down this route. If that’s the case, it will be just a matter of rephrasing this part of Figure 5a legend) and no discussion or analysis would be required.

Is there a reason why results depicted in Figure 5c,d,e,f are not tested using the spin null model? By doing that it will not be necessary to regress out distance (which as opposed to the spin approach it assumes a linear relationship)

Minor

Explicitly mention that the Lausanne dataset has been used elsewhere (e.g. Bazinet et al. 2021, Neuroimage).

Could authors briefly mention why deterministic rather than probabilistic tractography was used? I am afraid I do not understand the binning process of the DSI-based connectomes (“last paragraph of the Structural Network Reconstruction). Could authors rephrase this section? What is the objective of this preprocessing step? Is there any reference that readers can check?

The quantification of myelin content using MRI is still under debate. For example, to my knowledge, there are no articles correlating myelin histology with T1w/T2w ratio. For other approaches such as Magnetization Transfers, R2 ranges between 0.4 and 0.7 approximately. I understand that this is not within the scope of this manuscript (it doesn’t even show a significant association) but I recommend to the authors that they add a one or two lines outlining this limitation of MRI-based myelin markers (see Van der Weijden et al. 2021, Neuroimage, <https://www.sciencedirect.com/science/article/pii/S1053811920310466?via=ihub>, for an interesting discussion.).

I am intrigued that Schizophrenia and Schizotypy have such a differentiated influence of molecular vs. connectivity predictors. It is mentioned on the results but it would be interesting to mention on the discussion

Given that gene PC1 is likely to mainly reflect differences in cell type distribution, do authors consider that the association between gene PC1 and the thickness differences are driving by this factor?

I am happy to be approached by the authors if that could help with the revision.

Rafael Romero-Garcia (rromerog-ibis@us.es)

Reviewer #3 (Remarks to the Author):

In the manuscript "Molecular and connectomic vulnerability shape cross-disorder cortical abnormalities" the authors present an analysis of brain disorders. They consider data from the ENIGMA meta-consortium (N = 21,000 cases, N = 26,000 controls collected from over 50 studies in 40 countries; Thompson et al. 2020). They look at local phenotypes (such as gene expression, myelination) and global phenotypes (such as graph theoretical parameters such as centrality of the brain network). They find that local phenotypes are the best predictors of disease. They also find some interaction between local and global phenotypes (hence "molecular and connectomic" in the title). For example, they say "We consistently find that disorder-specific cortical abnormality is shaped more by the local molecular fingerprints of brain regions than network embedding." The conclusion that local effects are more important can also be seen clearly in Figure 3: most diseases (notably not schizophrenia) show the trend.

Their methodology is good, and this is some of the best evidence I have seen for the local hypothesis. They note that "How local attributes and global connectivity shape cross-disorder pathology remains an open question." This work provides evidence for the importance of local phenotypes over global phenotypes.

This work has the potential to influence priorities in neurology and molecular psychiatry (although, the authors can correct me if I'm wrong but I don't see anything immediately clinically relevant in this work).

Major points:

- Genetic sex has an impact on brain development. Should sex-stratified analysis be done? This may improve power by deconfounding, or lead to some further insight.
- The extent to which the global hypothesis is refuted by this work depends much on the quality of the global phenotypes. For example, if the noise level is higher in the global phenotypes, then they will be found to be less of a contributor to cortical abnormality even if a "noise free" version of the global phenotypes is actually more predictive. Can the authors somehow assess the noise level in the local - vs- global phenotypes (maybe with a simulation?) Or possibly repeat the construction of the global attributes using a different preprocessing pipeline?

What I mean is that 20 years from now, we'll have better technology for assessing graph properties of brains. How confident are the authors that their global evidence will still hold in 20 years, and is there anything more we can do now to gain confidence?

- This might be a good validation of the methodology: can the authors look at pairwise genetic correlation between the pathologies and conditions they study, and see if the pathologies and conditions that share genetic architecture significantly are also close together in the Figure 3 axis? This seems like a good control (towards gaining the sort of confidence I asked for in the previous point) and if the global attributes are robust they should emerge correlated in traits that have similar genetic etiology.

Minor points:

- Some short forms such as pd for Parkinson's and 22q for 22q11.2 deletion are inconsistently used (i.e., sometimes the short form is used othertimes not, which could be confusing).
- If it's possible, the title could be changed so that it's more clear that "shape" is a verb, for example "Cross-disorder cortical abnormalities are shaped by ..." I know that's longer, maybe there's another way. As it is, it is a bit hard to parse.
- On page 3, there is a typo "cortical abnormality" should be "cortical abnormality"

- For Figure 5: x-axis should be on same scale across all except panel a. Pearson's r in some panels looks like < -1 , is that because of jitter? panel b: The authors claim this is a normal distribution, can they provide a ks test? panel g: the top 2 and bottom 2 should likely be switched to match caption. Finally, significant correlation is not a contrast to low correlation. None of these are highly correlated (but, they are indeed all positively correlated).
- Typo in caption of Figure 1: "inhibitory".

Reviewer #4 (co-reviewed with reviewer 3 - no separate comments)

Dear Reviewers,

Thank you for the constructive feedback on our first submission, and for the opportunity to revise the manuscript. Following your comments and suggestions, we have thoroughly revised the manuscript. In this letter, we respond to each of the reviewers' comments in detail. Reviewer comments are in **bold font** and our responses are in regular font.

Major changes include: (a) repeating analyses using structural and functional connectomes from the Human Connectome Project (N=326), (b) outwardly addressing disease heterogeneity which includes a new analysis on age and disease severity, and (c) expanding on genetic and cognitive/behavioural perspectives on disease.

Reviewer #1 (Remarks to the Author):

“Molecular and connectomic vulnerability shape cross-disorder cortical abnormalities” by Hansen et al uses ENIGMA data to build a library of cortical thickness anomalies across 13 neurological and psychiatric disorders and compare these to molecular and connectivity fingerprints. The authors first ask whether each disorder, considered individually, aligns more closely to molecular gradients or regional connection topologies. They next build a transdiagnostic composite map and ask whether this has a stronger molecular or connectivity association.

The paper provides a novel approach to understanding transdiagnostic neurobiological perturbations to cortical grey matter, and their relationship to genetic and connectomic features. It provides a platform for future work to integrate other disorders or address developmental or ageing processes.

The paper is interesting, addresses a question of fundamental and clinical interest using advanced computational techniques. I think it would be of substantial interest to the readers of Nature Communications. I have mainly observations for consideration which the authors may choose to address with additional work, or through a deeper discussion of current limitations.

1. Each of the disorders is associated with a phenotypic profile (symptoms + neurocognitive disturbances). The study is somewhat limited for not incorporating these. For example, are there common/overlapping neurocognitive disturbances that correspond to the overlapping neurobiological changes. I assume ENIGMA has this information per disorder, that could be incorporated alongside the cortical thickness. This would extend the paper more deeply into translational territory and would be stronger than the superficial “reverse inference” observations on the top of page 8, second column. Likewise, are there symptom- or cognitive features that associate more strongly with the “molecular-correlated” disorders on Fig. 3 (ADHD, MDD – e.g. attention) versus those with a “connectivity” profile (SCZ – e.g. executive functioning). If this is to be deferred to future work, it should in the least be addressed in the Discussion.

We agree with the Reviewer that the phenotypic perspective of disease is missing. Unfortunately, the ENIGMA dataset does not include quantitative behavioural or cognitive tests. We are therefore unable to make statistical conclusions about how phenotypic profiles relate to molecular and connectomic contributions to disease, although we agree that this is an exciting future direction of this work. Nonetheless, we like the Reviewer's suggestion that molecular vs connectomic predictors may be related to a cognitive axis underlying disorder profiles (anchored by executive functioning and attention). We

have expanded on this point in the text (see also our response to Reviewer #3 Comment #3 who has a similar point about genetics).

(“Discussion” section, Paragraph #7):

“This work considers multimodal molecular and connectomic contributions to disorders but does not make conclusions about two important features of disease: cognitive phenotypes and genetics. An exciting future direction is to explore whether molecular and connectomic contributions to disease can be related to phenotypic or genetic similarity. Lee et al., 2019 compare single-nucleotide polymorphism data across eight psychiatric disorders and find that schizophrenia and bipolar disorder show greatest genetic similarity. This complements our finding that schizophrenia and bipolar disorder have consistent connectomic profiles. Lee et al., 2019 also find a clique among the disorders that we find are best predicted by molecular features: ADHD, autism, and major depressive disorder. On the other hand, a comprehensive battery of cognitive and behavioural tests was not uniformly applied to all the disease groups in the ENIGMA datasets. As a result, robust cross-disorder phenotypic profiles are less well established. Our findings potentially suggest an executive function (anchored by schizophrenia) versus attention (anchored by ADHD and ASD) cognitive axis that separates connectomic versus molecularly informed disorder profiles, but more work is needed to standardize cognitive testing and assess how cognitive/behavioural phenotypes may be related to brain structure. Altogether, future work is necessary to explore how overlapping genetic and neurocognitive disturbances correspond to molecular and connectomic contributions to disease.”

2. Most of the disorders – some more than others – are quite heterogenous over time (recent onset versus chronic), across age, and across patients (e.g. unipolar versus melancholic versus bipolar depression). Most of the patients in the clinical ENIGMA cohorts are quite stable and have been on pharmacological therapies, many of which have impacts on cortical thickness (e.g. anti-psychotics [1]). MDD, SCZ and ADHD in particular are not unitary entities across time or patients and their representation as singular entities in the various figures is a translational limitation. This approach is not at all unique to the present paper, but a widespread limitation so there is no line to be drawn in the sand here: Again, this could be addressed with some additional analyses (such as a break-out analysis of medication, age or time since illness onset analyses on at least one diagnostic category) or simply acknowledged in the Discussion

We thank the Reviewer for bringing up this point and agree that the heterogeneity of disorders deserves both additional analyses and discussion in the manuscript. We have included two new supplementary analyses that seek to track how well molecular/connectomic predictors map onto disease profiles across age and across disease severity. For a subset of four disorders (ADHD, bipolar disorder, depression, OCD), ENIGMA cortical abnormality maps exist for more age groups than only adults (specifically, pediatric and adolescent). We find that for ADHD, bipolar disorder, and depression, both molecular and connectomic predictions improve with age, whereas for OCD, predictions worsen with age (new figure after the next paragraph).

Next, we compare predictions (adjusted R^2) of Parkinson’s cortical abnormality profiles across patients at different stages of the disease. Here we find that molecular and connectomic predictors have mirroring effects across disease severity. Molecular predictors perform worse with disease severity, and

connectomic predictors perform better. This is exciting because it is in line with previous work showing how misfolded proteins in Parkinson's patients spread along the structural network.

We have added this analysis to the Results ("Results" section, "Local and global contributions to disorder-specific cortical morphology" subsection, Paragraph #4):

"One important consideration with this analysis is that disorder-specific pathology and symptom presentation are heterogeneous over time. The analysis in Figure 2 is limited to adults and encompasses multiple stages of disease progression. We therefore sought to investigate changes across different ages (pediatric, adolescent, and adult) and different disease severities. First, we tracked the model fit (adjusted R^2) of regression models that fit molecular/connectomic features to pediatric, adolescent, and adult cortical abnormality profiles for the four available disorders with this data (ADHD, bipolar disorder, depression, and OCD; Figure S4a). We find that model fit is greatest in adulthood, except for OCD which shows little change for connectomic predictors and a lower model fit in adulthood for molecular predictors. Next, focusing on disease severity, we show how model fit changes across four levels of Parkinson's disease severity (Hoehn and Yahr (HY) stages (Hoehn & Yahr, 1967); Figure S4b). Interestingly, we find that molecular predictors perform worse with disease severity whereas connectomic predictors perform better, supporting the notion that Parkinson's pathology is influenced by the spread of misfolded proteins on the structural connectome (Luk et al., 2012, Henderson et al. 2019, Zheng et al., 2019). Altogether, these analyses provide a more nuanced and transdiagnostic representation of molecular and connectomic contributions to cortical disorder vulnerability."

"Figure S4. Molecular and connectomic contributions to cortical abnormality change with age and disease severity | (a) Age stratified cortical abnormality patterns are available for ADHD, bipolar disorder, depression, and OCD. We fit molecular (darker line) and connectomic (fainter line) predictors to disorder profiles across different life stages. Note that the exact age range for each category differs slightly across the four disorders. (b) Cortical abnormality patterns were available for four stages of disease severity in Parkinson's disease, according to the Hoehn and Yahr (HY) stages (Hoehn & Yahr, 1967). We fit molecular (darker line) and connectomic (fainter line) predictors to cortical Parkinson's profiles at different stages of disease severity."

("Discussion" section, Paragraph #6; see also our response to Reviewer #2 Comment #9 and Reviewer #3 Comment #1):

"One strength of the ENIGMA consortium is that the datasets are pooled over thousands of individuals. However, such large-scale analyses obscure the important inter-subject variability that exists within all disorders. We conduct supplementary analyses in which cortical disorder profiles are stratified by age and disease severity (Figure S4) and find that molecular and connectomic contributions vary. For example, we find that molecular and connectomic influences on Parkinson's disease differ with disease severity: molecular predictors become less powerful predictors and connectomic predictors become more powerful predictors as the disease progresses. This complements previous work that suggests that atrophy in Parkinson's is the result of network-mediated spread of alpha-synuclein (Zheng et al., 2019). Furthermore, we find a similar trend of increased connectomic influence for severity of psychotic symptoms: namely, schizotypy and schizophrenia. Although schizotypy is not an earlier stage of schizophrenia (indeed, schizotypy is not a disorder per se but rather a multidimensional continuum of traits related to psychosis), individuals with schizotypy exhibit similar, albeit attenuated, characteristics as schizophrenia patients (Kirschner et al., 2021). We find that the connectomic influence is considerably greater in schizophrenia compared to schizotypy, which may suggest that the structural network gradually becomes more implicated in disease progression. One key factor that we were not able to study in more depth is that of biological sex. Since ENIGMA datasets are all sex-corrected, we are unable to make conclusions about how molecular and connectomic contributions may differ between the sexes. Multiple disorders show well-established sex-differences, including schizophrenia (Abel et al., 2010), autism (Werling & Geschwind, 2013), and depression (Altemus et al., 2014). Designing effective clinical interventions will require more nuanced studies that consider the many heterogeneities that exist within each disease."

Finally, we expand on the limitation that ENIGMA cohorts include medicated patients, including interventions that may have an effect on cortical thickness ("Discussion", Paragraph #8):

"The present work should be considered along with some important methodological considerations. First, although the ENIGMA consortium standardizes preprocessing pipelines and provides large N datasets, allowing for robust results and meaningful comparison between disorder-specific cortical abnormality maps, working with ENIGMA data also has caveats: (1) the measures of cortical abnormality are effect sizes between patients and controls and do not represent tissue volume loss/gain, (2) some of the patient populations included have co-morbidities and patients may be undergoing treatment, including treatment that may have an effect on cortical thickness (Voineskos et al., 2020), and (3) all analyses were conducted at the level of 68 cortical brain areas, limiting regional specificity and precluding analyses of the subcortex and cerebellum."

3. A methodological limitation is that the core method essentially derives from "correlations of correlations", limiting insights into deeper mechanistic causes. I think the existing "dig down" analyses are elucidating and largely overcome this weakness, but I would avoid statements like "regional molecular vulnerability and macroscale brain network architecture interact to drive the spatial patterning of cortical abnormalities in multiple disorders" which is something that could hardly have a counter-factual.

We have revised the manuscript to more clearly state the findings of the paper:

(“Abstract” section):

Old:

“We find that regional molecular vulnerability and macroscale brain network architecture interact to drive the spatial patterning of cortical abnormalities in multiple disorders”

New:

“We find a relationship between molecular vulnerability and white matter architecture that drives cortical disorder profiles.”

(“Introduction” section, Paragraph #4):

Old:

“Interestingly, for disorders that are better predicted by molecular attributes, we find that the spatial patterning of cortical abnormalities reflects the underlying network architecture, suggesting that the local molecular and global connectomic contributions to disorder effects may interact.”

New:

“Interestingly, for disorders that are better predicted by molecular attributes, we find that the spatial patterning of cortical abnormalities reflects the underlying network architecture, suggesting that the joint contribution of local molecular and global connectomic mechanisms is greater than their individual contribution.”

(“Results” section, “Interactions between local and global vulnerability” subsection, Paragraph #3):

Old:

“This finding is significant because it shows that connectome architecture interacts with local vulnerability.”

New:

“This finding is significant because it suggests that the combined effect of local vulnerability and connectome architecture is greater than their individual contribution.”

4. The main marker of neurobiological perturbation is cortical thickness: While this is a good choice, it is arguably not that surprising to see molecular associations stronger than connectomic ones, given that the former impact directly on cortical activity, function and integrity. But does intra-cortical pathology hold a privileged position when considering the pathogenesis of brain disorders? Some might argue that many disorders arise primarily in white matter connections, and that these then propagate to cortical grey matter. This is a relatively minor point but might warrant a brief note in the paper.

We agree with the Reviewer that white matter architecture likely plays an important role in disease progression. In fact, one of our key findings (Figure 3) is that disorders that are well predicted by molecular predictors (i.e. have large R^2_{adj}) also show the “network spreading” phenomenon whereby cortical abnormality profiles are organized such that regions with high abnormality are connected with one another. This suggests that the connectome may play a role in “projecting” local abnormalities (i.e.

aberrant gene expression, misfolded proteins, etc.) to connected brain regions. These local abnormalities may originate in the cell bodies but may also originate from the axons between neurons.

We have modified the text to expand on this point (“Discussion” section, Paragraph #4):

“We generally find that cortical abnormality is better predicted by local vulnerability compared to global connectomic vulnerability. One possible reason for the relatively poorer performance of connectivity predictors is that they are generic measures of a region’s embedding in a network (number of connections, centrality, connection diversion) but do not consider how this embedding exposes regions to pathology elsewhere in the network. Indeed, we find that disorders whose cortical morphology is better reflected by local vulnerability, the abnormality pattern is organized such that areas with greater abnormality are disproportionately more likely to be structurally- and functionally-connected with each other (e.g. ASD, ADHD, 22q11.2 deletion syndrome, temporal lobe epilepsy, schizotypy, bipolar disorder). This suggests a network spreading phenomenon where focal pathology or perturbation propagates to connected regions, resulting in cortical abnormality that is correlated with the underlying connection patterns (Fornito, 2015). This interaction between local vulnerability and connectomic vulnerability has previously been reported in neurodegenerative syndromes where the trans-synaptic spreading of misfolded proteins appears to be guided and amplified by local gene expression (Cornblath et al., 2021, Henderson et al., 2019, Raj et al., 2021, Zheng et al., 2019, Shafiei & Bazinet et al., 2022). In other words, the poorer performance of connectivity predictors does not suggest that the white matter architecture is less relevant to disease progression. Indeed, pathogenesis of multiple diseases is thought to originate in the white matter of the brain (Fornito et al., 2015, Bartzokis 2011, Hirokawa et al., 2010). A promising future direction for studying cross-disorder brain abnormalities is to focus on disruptions in white-matter pathways instead of cortical thickness (de Lange et al., 2019, Binette et al., 2021).”

Minor:

5. I thought it was only the analysis pipelines that were harmonized across ENIGMA not the image acquisition (p9) which were quite site specific (but I could be wrong!).

We thank the Reviewer for catching this error, as ENIGMA data is only harmonized across processing and analysis pipelines, not imaging. We have corrected the error (“Methods” section, “Cortical disorder maps” subsection):

“The ENIGMA (Enhancing Neuroimaging Genetics through Meta-Analysis) Consortium is a data-sharing initiative that relies on standardized processing and analysis pipelines, such that disorder maps are comparable (Thompson 2020).”

6. The overall (anti-)correlations among the graph metrics are stronger than the molecular features (heatmap of Figure 1), which likely has an impact on how the relative and adjusted R² values are derived using the dominance analysis (as the effective degrees of freedom differ and there does not seem to be a step to partial out mutual correlations).

We agree with the Reviewer that the structure of the molecular and connectomic predictors will have an influence on the overall result. We have conducted two supplementary analyses to compare the

correlational structure of the molecular and connectomic predictor sets. First, we plot the distribution of absolute correlation coefficients from the heatmap in Figure 1. Although the connectomic predictors have greater absolute correlations than molecular predictors, this difference is not statistically significant (Welch's t-test, $p=0.11$). Second, to provide readers more insight into the correlation structure of each predictor set, we apply PCA to both predictor sets (regions x predictor matrices) and plot the percent variance explained for each component. Here we find that the first principal component of connectomic predictors explains more variance (61%) than the first principal component of the molecular predictors (47%), consistent with the idea that the connectomic predictors tend to have more prominent correlational structure. We have included this figure in the supplement to be transparent about the correlational structure of both predictor sets (Figure S15).

“Figure S15. Comparing molecular and connectomic predictor sets | (a) Distributions of absolute correlations (Pearson’s r) between pairs of predictors are not significantly different from one another (Welch’s t-test, $p=0.11$). (b) PCA was applied to the region x predictor matrices for molecular and connectomic predictors separately. The first principal component of connectomic predictors explains more variance (61%) than that of molecular predictors (47%).”

Although we try to be comprehensive for both the molecular and connectomic perspectives, these results are naturally contingent on the predictor subsets chosen. We decided not to partial out any mutual correlations from the predictors because we wanted to keep any relationships between input variables. Nonetheless, this mismatch in correlational structure of the predictor sets is certainly a limitation to the study. We have therefore expanded on this point in the text (“Discussion” section, Paragraph #8):

“Fourth, we assessed the contribution of multiple predictors to disorder maps using simple but robust linear models that are not sensitive to non-linear contributions or higher-order interactions among the predictors. In addition to this, the correlational structure of the predictor subsets affects predictive power which limits our ability to compare molecular and connectomic model fits (Figure S15).”

7. p4: Why “embryonic” stage. Wiring abnormalities can arise across any neurodevelopmental stage.

We have updated the text accordingly (“Results” section, “Interactions between local and global vulnerability” subsection, Paragraph #1):

“In neurodegenerative diseases, this interaction may result in synaptic pruning and cortical atrophy whereas in neurodevelopmental disorders, the pathology may manifest as perturbations in network wiring during development (Di Martino et al., 2014).”

8. The MRI data used to reconstruct the normative functional and structural connectivity data is not particularly state of the art (long TR, 32 channel head coil, single-shell, deterministic tractography etc). I doubt this will impact on the bottom-line findings, but I am not sure why the authors did not choose a more advanced HCP-like data set to work with.

We now repeat all analyses using structural and functional data from HCP (N=326), and find consistent results. Results from the original dataset are shown in the main text and results from the HCP dataset are shown in the supplement. Encouragingly, the two datasets show consistent results.

Figure SX:

“Figure S11. Replication using HCP structural networks | The main analysis (corresponding to Figure 2 in the main text) was repeated using diffusion weighted MRI data from the Human Connectome Project (N=326) (Van Essen et al., 2013).”

9. The authors note the limitations of the 68-region DK cortical atlas – and the rationale for using it. But the regions in this atlas differ highly in extent and it would be worth checking that regions

that dominate the cross-diagnostic findings are not simply larger in extent and therefore more likely to overlap by chance, and more likely to have higher SnR for all the analysis steps.

We have included a supplementary figure to the revised text to show the correlation between cortical abnormality profile and size of Desikan-Killiany region (defined as the number of MNI152 voxels within a region), and we find that there is not a consistent influence on parcel size and cortical abnormality.

“Figure S13. Effects of parcel size on disorder profiles | Each cortical abnormality disorder profile (y-axis) was compared to the region size of each parcel in the Desikan Killiany 68-region atlas (x-axis). Parcel size was defined as the number of MNI152 voxels in each parcel.”

We also compare the epicentre likelihood maps with parcel size (which is only done for the seven disorders that show significance correlation between node and mean neighbour abnormality). We don't find a consistent relationship between epicentre likelihood and distance although we do find positive correlations for the epilepsies and for bipolar disorder.

“Figure S14. Distance and parcel size effects on epicentre likelihood | For the seven disorders that show significant correlation between node and sc-/fc-weighted neighbour abnormality, we correlated epicentre likelihood with (a) the average Euclidean distance

between one brain region and all other brain regions, and (b) the parcel size (defined as number of voxels defined in the MNI-152 atlas) of each brain region.”

(“Results” section, “Sensitivity and robustness analyses” subsection, Paragraph #2):

“Next, since the Desikan-Killiany atlas parcellates the brain into unequally sized parcels, we tested the effect of parcel size on disorder abnormality maps. Parcel size was defined as the number of voxels assigned to the parcel using the MNI-152 volumetric parcellation. Across all thirteen disorder maps, we do not find a significant correlation between parcel size and cortical abnormality (Figure S13). Likewise, we compare effects of parcel size on epicentre likelihood maps (Figure S14b). We find no significant correlations except between parcel size and bipolar epicentre likelihood ($r=0.44$, $p_{\text{spin}}=0.03$).”

10. A point perhaps for another occasion, but I am a bit sceptical about the now widely used spin test to represent nulls for spatial correlations between data sets: While it is crucial that spatial correlations be preserved in nulls, the process of conflating and then collapsing the cortical surface will introduce geometric distortions that over-estimate the true underlying null variance: spatial correlations from highly curved regions will be compressed onto the inflated representation which are then spun and mapped onto flatter regions. While it seems likely that spatial correlations will be preserved ‘on average’ the variance (the width of the null distribution) will be a composite of the true variance plus these geometric effects. A more accurate null could be obtained through whitening then resampling on the naïve cortical geometry. I do not think this is an issue for the present paper (because if anything a wider null means a more conservative test), but hope this limitation either be rebutted or acknowledged in the corresponding Methods section of the ms (p12).

This is an interesting point that we had not previously considered. Although the Reviewer mentions it is a minor point, we were interested in finding out whether spin-rotated null distributions might be too wide. To do this, we used a generative framework (a variogram model originally introduced by Burt et al., 2020) to make surrogate null maps that retain spatial features characteristic of the data from which they are estimated. These spatial null models don’t involve spins (i.e. conflating and collapsing the cortical surface) but also don’t reproduce the spatial autocorrelation of the data as well (Markello & Misic, 2021).

To test this, we used the ADHD cortical abnormality map and its correlation with mean neighbour abnormality (scatterplot shown in Figure S5 and results used in Figure 3 of the main text). We generate 10,000 rotations of the ADHD cortical abnormality map (using the spin-test) as well as 10,000 generative null models (using the Burt et al., 2020 generative method). Interesting, as the Reviewer predicted, we find that the generative null model has a narrower distribution that would result in smaller p-values.

One concern that we have with this parameterized method is that generative models have a harder time reproducing the spatial autocorrelation of the data. Here we show the Moran's I (a metric of the degree of spatial autocorrelation) for the empirical data (vertical line) compared to the 10,000 spin and generative nulls. Indeed, the spin null better reproduces the autocorrelational structure of the data. One reason the generative models may struggle is that the parcellation is coarse and composed of uneven parcel sizes.

Additionally, we can't apply generative nulls to correlations between similarity matrices (as is done in Figure 5) because the data is not regional.

For these reasons, we have opted to keep the spin-permuted nulls. We thank the Reviewer for this comment which sparked this exploration and a better understanding of how spin and generative nulls behave. We mention this potential limitation in the text ("Methods" section, "Null models" subsection, Paragraph #1):

"This spin-permuted null model involves conflating and collapsing the brain surface to and from a sphere. The geometry of the cortical surface is therefore not retained in the spinning process, which may result in null distributions that are too wide."

Burt, J. B., Helmer, M., Shinn, M., Anticevic, A., & Murray, J. D. (2020). Generative modeling of brain maps with spatial autocorrelation. *NeuroImage*, 220, 117038.

Markello, R. D., & Misic, B. (2021). Comparing spatial null models for brain maps. *NeuroImage*, 236, 118052.

References:

1. Voineskos, A. N., Mulsant, B. H., Dickie, E. W., Neufeld, N. H., Rothschild, A. J., Whyte, E. M., ... & Flint, A. J. (2020). Effects of antipsychotic medication on brain structure in patients with major depressive disorder and psychotic features: neuroimaging findings in the context of a randomized placebo-controlled clinical trial. *JAMA psychiatry*, 77(7), 674-683.

Reviewer #2 (Remarks to the Author):

This manuscript is a rigorous and methodologically sound study that addresses a key neuroscience question. The authors present a relevant set of results that will be of major interest to the research community. Evidently, one of the strongest aspects of the study is the data that has been exploited. Although this (ENIGMA) data has been extensively analysed before in previous studies (cited by the authors), the exhaustive coverage of a wide range of local (molecular) and global (connectivity) metrics using other public datasets demonstrates how integrative approaches can provide new insights on the molecular underpinnings of structural brain alterations associated to brain disease. First of all, I want to congratulate the authors for the important and valuable investment that they have made with the Code and Data repository. The null models and statistical procedures are appropriate (spin test, cross-validations, etc.). Guidelines are clear, the code is commented and there is an honest effort to facilitate the replicability of their findings. Despite the large number of results, they are all clearly explained and detailed. The schematics are very illustrative and the narrative is easy to follow. I have some comments that I hope can contribute to improving the manuscript:

1) For some analyses, a considerable number of tests have been carried out (for example those depicted in Figure 2). I understand that correcting for multiple comparisons could be problematic here (metrics are not independent) but, if no correction procedure is used I think it should be at least acknowledged on the limitation section.

This is an important point - dominance p-values were not corrected for multiple comparisons. The dominance analysis is intended as an informative and interpretable visualization of how the fit (adjusted R^2) of each model is distributed across molecular/connectomic features. We initially applied the spin-test to the dominance of each input variable to provide an extra source of information, but ultimately, following the Reviewer's comment, we think that this additional significance testing is superfluous and potentially misleading.

One advantage of dominance analysis is that this comparison between input variables is straightforward: if the dominance of variable X_1 is twice that of variable X_2 , the interpretation is that X_1 contributes twice as much as X_2 to the prediction of Y . Therefore, dominances should be interpreted with respect to other input variables of the same model. Adding p-values gives the incorrect impression that an input variable acts in isolation.

We have therefore removed the significance testing from the dominance analysis. This does not affect the results nor the conclusions of the study, as the specific dominance analysis results play a minor part in the manuscript and those that were labeled significant are also those with large magnitude.

We clarify this in the revised manuscript ("Methods" section, "Dominance analysis" subsection):

"Dominance analysis seeks to determine the relative contribution ("dominance") of each input variable to the overall fit (adjusted R^2) of the multiple linear regression model (<https://github.com/dominance-analysis/dominance-analysis> (Azen & Budescu, 2003, Budescu 1993). This is done by fitting the same regression model on every combination of input variables (2^p-1 submodels for a model with p input variables). Total dominance is defined as the average of the relative increase in R^2 when adding a single input variable of interest to a submodel, across all 2^p-1 submodels. The sum of the dominance of all input variables is equal to the total adjusted R^2 of the complete model, making total

dominance an intuitive measure of contribution. Note that significance testing is not applied to the individual dominances because the contributions of input variables are relative to other predictors in the model and input variables do not act in isolation.”

Figure 2:

a | multilinear models

b | dominance analysis

2) Is the epicentre likelihood related to Euclidean distance? I found intriguing that the highest epicentre likelihood is located at motor (and parietal) cortex. I would have expected that association regions and hubs will constitute these epicentres. As authors point out “In particular, regions that are highly connected and potentially important for communication tend to be disproportionately affected by disease [26, 114].”. I wonder whether I am misinterpreting the metric or whether the distance effect on the connectivity matrix influences the epicentre likelihood. Interestingly, authors discuss this finding in the discussion but arguments are focused on previous evidence showing that sensory-motor cortex is a plausible epicentre. Could you please discuss whether this is specific to primary cortex? or rather there is also evidence of the opposite (epicentres predominantly in associations cortices). Note: Looking at Figure S7, I think that one possibility is that this is driven by the epilepsy (which is duplicated). Given that these maps show considerable differences in regional distribution across conditions, is the average a meaningful metric?

To test whether epicentre likelihood is related to distance, we correlated the epicentre likelihood of the seven disorders to the average Euclidean distance between a region and all other regions. 22q11.2 deletion syndrome and ADHD do show moderate negative correlations with distance (i.e. central regions have greater epicentre likelihood), but the remaining disorders show no relationship (except bipolar disorder which oddly shows a positive correlation ($r=0.37$) such that likely epicentres are actually farther away from other brain regions). In other words, we don't find consistent evidence for epicentre likelihood being distance-driven.

a | distance effects

“Figure S14. Distance and parcel size effects on epicentre likelihood | For the seven disorders that show significant correlation between node and sc-/fc-weighted neighbour abnormality, we correlated epicentre likelihood with (a) the average Euclidean distance between one brain region and all other brain regions, and (b) the parcel size (defined as number of voxels defined in the MNI152 atlas) of each brain region.”

(“Results” section, “Sensitivity and robustness analyses” subsection, Paragraph #2):

“Finally, since epicentre likelihood is calculated using the structural connectome, we also assessed the relationship between epicentre likelihood and distance. Specifically, we correlate epicentre likelihood with the average distance between a brain region and all other brain regions (Figure S14a). We do not find any significant correlations between epicentre likelihood and distance.”

We agree with the Reviewer that the average epicentre likelihood might not be the most meaningful metric, especially considering that both right and left temporal lobe epilepsy have similar epicentre likelihood maps (and may be biasing the average). We therefore combine the left and right temporal lobe epilepsy epicentre likelihood maps into a single average. Then, for these six disorders (five disorder maps plus the composite epilepsy epicentre map), we look at (1) the mean epicentre likelihood, (2) the median epicentre likelihood, and (3) a map where each brain region is coloured by the number of disorders for which the brain region is in the top 50% of most likely epicentres.

“Figure S8. Cross-disorder epicentre likelihood maps | Epicentre likelihood was calculated for the seven disorders that show significant correlation between node and mean neighbour abnormality. To avoid biasing results, right and left temporal lobe epilepsy epicentre likelihood is represented by a single mean likelihood map (see disorder-specific epicentre likelihood maps in Figure S7). Epicentre likelihood can be calculated as the (a) mean likelihood, (b) median likelihood (shown in Figure 4c), and (c) frequency a brain region is in the top 50% of most likely epicentres across the included disorders.”

We find similar results across cross-disorder epicentre likelihood maps, including the presence of inferior temporal cortex, posterior parietal cortex (especially the angular gyrus), and primary motor cortex (especially in the left hemisphere). Bilaterally, the precuneus and superior parietal cortex appear as cross-disorder epicentres. We choose to show the median epicentre likelihood map in the main text because it is a continuous map (unlike the frequency map) and is not as easily skewed by single disorders (unlike the mean map), but we note that the mean map and the frequency map are both viable alternatives.

(“Results” section, “Interactions between local and global vulnerability” subsection, Paragraph #4):

“Next, we aimed to construct a single cross-disorder epicentre likelihood map (Figure 4c). To avoid having left and right temporal lobe epilepsy - which demonstrate similar epicentre likelihood maps - bias the cross-disorder likelihood map, we combined left and right epilepsy epicentre likelihood into a single average map. We calculated the median epicentre likelihood across these six disorders and find that cross-disorder epicentre likelihood is highest in bilateral sensory-motor cortex, medial temporal lobe, precuneus, and superior parietal cortex. In Figure S8 we show mean epicentre likelihood as well as a map that shows the frequency with which a brain region is in the top 50% of most likely epicentres across the six disorders. Across all three methods (mean, median, frequency), cross-disorder epicentre likelihood is consistent.”

Since the cross-disorder epicentres include both unimodal and transmodal brain regions, we discuss both the perspective that the sensory-motor cortex is an epicentre as well as the perspective that transmodal regions are epicentres.

(“Discussion” section, Paragraph #5):

“The interaction between molecular vulnerability and network structure naturally raises the question of what are the network epicentres of cortical disorder maps. We find cross-disorder epicentres - regions with high abnormality that are also strongly connected with other regions with high abnormality - in primarily transmodal regions (e.g. inferior temporal lobe, angular gyrus, precuneus, superior parietal cortex), although the motor cortex also appears as an epicentre. That unimodal regions such as the motor cortex is an epicentre is consistent with recent reports that multiple psychiatric and neurological disorders are accompanied by sensory deficits and reduced motor control (Marco et al., 2011, Javitt et al., 2015, Bernard et al., 2015). Indeed, the sensory-motor cortex has been previously established as a functional hub in temporal lobe epilepsies and across multiple psychiatric disorders (Lariviere et al., 2020, Kebets et al., 2019). Interestingly, the precuneus and superior parietal cortex are members of the brain's putative rich club - densely inter-connected regions that are thought to support the integration and broadcasting of signals (van den Heuvel & Sporns., 2011). Rich club regions undergo changes in connectivity patterns in multiple diseases such as schizophrenia, Alzheimer's, and Huntington's (van den Heuvel et al., 2013, van den Heuvel & Sporns, 2019, deLange et al, 2019, Crossley et al., 2014, McColgan et al., 2015). We complement this work by showing that the precuneus and superior parietal cortex are both vulnerable to cortical abnormality and, by virtue of their network embedding, increase disease exposure to connected regions. Conversely, although the anterior cingulate cortex (ACC) is implicated across multiple psychiatric disorders (Shafiei et al, 2020, Goodkind et al., 2015), we do not find that the ACC is an epicentre of cross-disorder cortical morphology. This suggests that although the ACC demonstrates considerable local vulnerability in a subset of brain disorders, it is not consistently involved across the seven disorders included in the epicentre analyses. Altogether, despite variable cortical morphology patterns across the thirteen disorders, when looked at through the lens of network connectivity, we see a more consistent and compact subset of potential epicentres, suggesting greater commonality among diseases than previously appreciated.”

3) The disorder similarity matrix shown in Figure 5a looks very similar to a Structural Covariance Network. Given that it is computed as “pairwise correlation of regional cortical thickness” (according to figure 5a legend) I wonder whether the associations described later are structural-Covariance-related (e.g. as described in Romero-Garcia et al. 2018, Neuroimage) rather than disorder-related. Would it make sense to compute the disorder similarity as the correlation between “cortical thinning (for example, the difference with controls or the T-scores)” rather than using raw cortical thickness? Note: The Disorder similarity methods section states that “we correlated the abnormality vectors” so authors may already have gone down this route. If that's the case, it will be just a matter of rephrasing this part of Figure 5a legend) and no discussion or analysis would be required.

We apologize for our inconsistent description of the disorder similarity matrix. Disorder similarity was calculated by correlating vectors of the cortical abnormality measure, which is the case vs. control “thinning” metric shown on the brain surfaces in Figure 2. In other words, disorder similarity was not calculated using raw cortical thickness. This was a poor choice of words on our part and has been fixed in the manuscript.

This point has also inspired us to contextualize disorder similarity within the broader literature on annotation similarity (“Results” section, “Brain regions with similar molecular annotations are similarly affected across disorders” subsection, Paragraph #1):

“In the previous sections, we mapped molecular annotations and network measures to each disorder separately. Here, we focused on disorder similarity. For every region we constructed a 13-element vector of abnormality values, where each element corresponds to cortical change (i.e. cortical abnormality) in that region in one disorder. We then correlated regional vectors with each other to estimate how similarly two regions are affected across the thirteen disorders (Figure 5a). Disorder similarity is analogous to other measures of inter-regional attribute similarity including anatomical covariance (Romero-Garcia et al., 2018, Evans et al., 2013, Vasa et al., 2018), morphometric similarity (Seidlitz et al., 2018), correlated gene expression (Arnatkeviciute et al., 2021, Fulcher et al., 2016, Richiardi et al., 2015), receptor similarity (Hansen et al., 2021), temporal profile similarity (Shafiei et al., 2020), and microstructural similarity (Paquola et al., 2019).”

4) Is there a reason why results depicted in Figure 5c,d,e,f are not tested using the spin null model? By doing that it will not be necessary to regress out distance (which as opposed to the spin approach it assumes a linear relationship)

We have revised the analysis and now apply the spin null to the correlations between matrices. This is done by spinning both rows and columns in the similarity matrices (region x region disorder similarity, molecular similarity, connectivity similarity, receptor similarity, correlated gene expression, functional connectivity). The new Figure 5 now depicts the correlations between matrices without distance regression using the spin null, and we have moved the distance-regressed versions to the Supplement (Figure S9).

Minor

5) Explicitly mention that the Lausanne dataset has been used elsewhere (e.g. Bazinet et al. 2021, Neuroimage).

We now mention that the Lausanne dataset has been used elsewhere (“Methods” section, “Structural and Functional networks” subsection):

“The Lausanne dataset is available at <https://zenodo.org/record/2872624#.XOJqE99fhmM> and has been used in other work (Vazquez-Rodriguez et al., 2019, Bazinet et al., 2021).”

Vázquez-Rodríguez, B., Suárez, L. E., Markello, R. D., Shafiei, G., Paquola, C., Hagmann, P., ... & Masic, B. (2019). Gradients of structure–function tethering across neocortex. *Proceedings of the National Academy of Sciences*, 116(42), 21219-21227.

Bazinet, V., de Wael, R. V., Hagmann, P., Bernhardt, B. C., & Masic, B. (2021). Multiscale communication in cortico-cortical networks. *NeuroImage*, 243, 118546.

6) Could authors briefly mention why deterministic rather than probabilistic tractography was used?

We now show results using both deterministic (the original Lausanne dataset) and probabilistic (HCP dataset) tractography (see also our response to Reviewer #1 Comment #8 and Reviewer #3 Comment #2). Results using the original Lausanne dataset are in the main text and results using the HCP dataset are in the supplement. We find consistent results across both datasets.

a | multilinear models

b | dominance analysis

“Figure S11. Replication using HCP structural networks | The main analysis (corresponding to Figure 2 in the main text) was repeated using diffusion weighted MRI data from the Human Connectome Project (N=326) (Van Essen et al., 2013).”

7) I am afraid I do not understand the binning process of the DSI-based connectomes (“last paragraph of the Structural Network Reconstruction). Could authors rephrase this section? What is the objective of this preprocessing step? Is there any reference that readers can check?

The binning process applied to the structural connectomes is part of the procedure to create a single consensus network that preserves the density and edge-length distributions of individual connectomes. This method was first used in Misić et al., 2015 but is formally presented in Betzel et al., 2019 where the method is compared to other group-representative network construction methods.

We have revised the text to clarify the motivation and procedure, as well as including the relevant references (“Methods” section, “Group-consensus structural network” subsection):

“To construct a group-level connectome, we used a consensus approach that seeks to preserve the density and edge-length distributions of the individual connectomes (first applied in Misić et al., 2015 and presented formally in Betzel et al., 2019). This procedure better captures important organizational features of subject-level networks compared to other consensus methods (i.e. thresholding based on whether an edge is observed in a

fraction of subjects) (Betzel et al., 2019). The procedure for generating the consensus network is as follows. First, existing edges across participants were binned according to length. The number of bins was determined heuristically as the square root of the mean binary density across participants. Within each bin, the k most frequently occurring edges across participants were retained. k was set as the average across the number of edges each participant has in the bin. To ensure that interhemispheric edges are not underrepresented, we carried out this procedure separately for inter- and intrahemispheric edges.”

Mišić, B., Betzel, R. F., Nematzadeh, A., Goni, J., Griffa, A., Hagmann, P., ... & Sporns, O. (2015). Cooperative and competitive spreading dynamics on the human connectome. *Neuron*, 86(6), 1518-1529.

Betzel, R. F., Griffa, A., Hagmann, P., & Mišić, B. (2019). Distance-dependent consensus thresholds for generating group-representative structural brain networks. *Network neuroscience*, 3(2), 475-496.

8) The quantification of myelin content using MRI is still under debate. For example, to my knowledge, there are no articles correlating myelin histology with T1w/T2w ratio. For other approaches such as Magnetization Transfers, R2 ranges between 0.4 and 0.7 approximately. I understand that this is not within the scope of this manuscript (it doesn't even show a significant association) but I recommend to the authors that they add a one or two lines outlining this limitation of MRI-based myelin markers (see Van der Weijden et al. 2021, Neuroimage, <https://www.sciencedirect.com/science/article/pii/S1053811920310466?via=ihub>, for an interesting discussion.).

We have revised the manuscript to address the limitation of using the T1w/T2w ratio as a proxy for myelin content.

(“Methods” section, “Molecular predictors” subsection):

“*Myelination*. Data from the Human Connectome Project (HCP, S1200 release; Van Essen et al., 2013, Glasser et al., 2013) was used for measures of T1w/T2w ratios—a proxy for intracortical myelin—for 417 unrelated participants (age range 22–37 years, 193 males), as approved by the WU-Minn HCP Consortium. Images were acquired on a Siemens Skyra 3T scanner, and included a T1-weighted MPRAGE sequence at an isotropic resolution of 0.7mm, and a T2-weighted SPACE also at an isotropic resolution of 0.7mm. Details on imaging protocols and procedures are available at <http://protocols.humanconnectome.org/HCP/3T/imaging-protocols.html>. Image processing includes correcting for gradient distortion caused by non-linearities, correcting for bias field distortions, and registering the images to a standard reference space. T1w/T2w ratios for each participant was made available in the surface-based CIFTI file format and parcellated into 68 cortical regions according to the Lausanne anatomical atlas (Cammoun et al., 2012). Note that the T1w/T2w ratio is an MRI-based estimate of myelin content that has not yet been validated against myelin histology (Van der Weijden et al., 2021). Other MRI-based proxies may be more suitable alternatives, such as magnetization transfer or simultaneous tissue relaxometry of R_1 and R_2 relaxation rates and proton density (SyMRI), which have been validated using myelin histology and are closely correlated to one another (Hagiwara et al., 2018, Van der Weijden et al., 2021). Additionally, PET imaging may be a promising avenue for mapping myelin content in the brain (Auvity et al., 2020, Zeydan et al., 2018).”

("Discussion" section, Paragraph #8):

"Additionally, the molecular predictors are limited by imaging modality (in particular myelination, for which the T1w/T2w ratio is an indirect estimate (Van der Weijden et al., 2021, Hagiwara et al., 2018), and, in the case of the gene and receptor gradients, by the subset of genes and receptors included in the data decomposition."

9) I am intrigued that Schizophrenia and Schizotypy have such a differentiated influence of molecular vs. connectivity predictors. It is mentioned on the results but it would be interesting to mention on the discussion

We have added a new paragraph about how the influence of molecular and connectomic predictors changes with disease severity (as well as age and sex). This paragraph complements a new analysis on PD disease severity (prompted by Reviewer #1 Comment #2), where we find that connectomic predictors become more important with disease severity. Likewise, we find that the connectomic predictors perform better in schizophrenia than schizotypy.

We discuss this in the text (see also our response to Reviewer #1 Comment #2 and Reviewer #3 Comment #1; "Discussion" section, Paragraph #6):

"One strength of the ENIGMA consortium is that the datasets are pooled over thousands of individuals. However, such large-scale analyses obscure the important inter-subject variability that exists within all disorders. We conduct supplementary analyses in which cortical disorder profiles are stratified by age and disease severity (Figure S4) and find that molecular and connectomic contributions vary. For example, we find that molecular and connectomic influences on Parkinson's disease differ with disease severity: molecular predictors become less powerful predictors and connectomic predictors become more powerful predictors as the disease progresses. This complements previous work that suggests that atrophy in Parkinson's is the result of network-mediated spread of alpha-synuclein (Luk et al., 2012, Henderson et al. 2019, Zheng et al., 2019). Furthermore, we find a similar trend of increased connectomic influence for severity of psychotic symptoms: namely, schizotypy and schizophrenia. Although schizotypy is not an earlier stage of schizophrenia (indeed, schizotypy is not a disorder per se but rather a multidimensional continuum of traits related to psychosis), individuals with schizotypy exhibit similar, albeit attenuated, characteristics as schizophrenia patients (Kirschner et al., 2021). We find that the connectomic influence is considerably greater in schizophrenia compared to schizotypy, which may suggest that the structural network gradually becomes more implicated in disease progression."

10) Given that gene PC1 is likely to mainly reflect differences in cell type distribution, do authors consider that the association between gene PC1 and the thickness differences are driven by this factor?

We agree with the Reviewer that the genomic gradient is likely reflective of cell type distribution. The original manuscript did not properly contextualize the genomic gradient and why it is relevant. We have now emphasized the relevance of the genomic gradient.

("Methods" section, "Biological predictors" subsection, Paragraph #2):

“The first principal component of gene expression (“gene gradient”) was used to represent the variation in gene expression levels across the left cortex. This gradient has been previously related to cell type distributions and cell-specific gene expression, which suggests the gradient is related to the cellular architecture of the brain (Hawrylycz et al., 2012, Burt et al., 2018, Seidlitz et al., 2020, Anderson et al., 2020, Hansen et al., 2021).”

(“Results” section, Paragraph #1):

“The molecular fingerprint of a region was defined using the gene expression gradient (a potential proxy for cell type distribution (Hawrylycz et al., 2012, Burt et al., 2018, Seidlitz et al., 2020, Anderson et al., 2020, Hansen et al., 2021)), neurotransmitter receptor gradient, excitatory-inhibitory receptor density ratio, glycolytic index, glucose metabolism, synapse density, and myelination.”

(“Results” section, “Local and global contributions to disorder-specific cortical morphology” subsection, Paragraph #3):

“From the dominance analysis, we find that certain predictors are consistently unimportant. Indeed, synapse density and myelination demonstrate less dominance than microscale gradients such as the gene expression gradient (a potential proxy for cell type distribution (Hawrylycz et al., 2012, Burt et al., 2018, Seidlitz et al., 2020, Anderson et al., 2020, Hansen et al., 2021)), neurotransmitter receptor gradient, and metabolic gradients.”

I am happy to be approached by the authors if that could help with the revision.

Rafael Romero-Garcia (rromerog-ibis@us.es)

Reviewer #3 (Remarks to the Author):

In the manuscript "Molecular and connectomic vulnerability shape cross-disorder cortical abnormalities" the authors present an analysis of brain disorders. They consider data from the ENIGMA meta-consortium (N = 21,000 cases, N = 26,000 controls collected from over 50 studies in 40 countries; Thompson et al. 2020). They look at local phenotypes (such as gene expression, myelination) and global phenotypes (such as graph theoretical parameters such as centrality of the brain network). They find that local phenotypes are the best predictors of disease. They also find some interaction between local and global phenotypes (hence "molecular and connectomic" in the title). For example, they say "We consistently find that disorder-specific cortical abnormality is shaped more by the local molecular fingerprints of brain regions than network embedding." The conclusion that local effects are more important can also be seen clearly in Figure 3: most diseases (notably not schizophrenia) show the trend.

Their methodology is good, and this is some of the best evidence I have seen for the local hypothesis. They note that "How local attributes and global connectivity shape cross-disorder pathology remains an open question." This work provides evidence for the importance of local phenotypes over global phenotypes.

This work has the potential to influence priorities in neurology and molecular psychiatry (although, the authors can correct me if I'm wrong but I don't see anything immediately clinically relevant in this work).

Major points:

1) Genetic sex has an impact on brain development. Should sex-stratified analysis be done? This may improve power by deconfounding, or lead to some further insight.

We agree with the Reviewer that sex has an effect of disease pathology and, especially from a clinical perspective, is an important consideration to take into account. Unfortunately, the ENIGMA data used in the present work are not stratified by sex. Instead, all ENIGMA-derived cortical abnormality maps (i.e. case versus control cortical thickness) were derived after age and sex correction. Therefore, our analyses make conclusions on how local and global features map onto disorder profiles outside of the effects of sex.

Nonetheless, we agree that a sex-stratified analysis is of interest and therefore discuss the effects of sex on disease in the Discussion (see also our response to Reviewer #1 Comment #2 and Reviewer #2 Comment #9).

("Discussion" section, Paragraph #6):

"One strength of the ENIGMA consortium is that the datasets are pooled over thousands of individuals. However, such large-scale analyses obscure the important inter-subject variability that exists within all disorders. We conduct supplementary analyses in which cortical disorder profiles are stratified by age and disease severity (Figure S4) and find that molecular and connectomic contributions vary. For example, we find that molecular and connectomic influences on Parkinson's disease differ with disease severity: molecular predictors become less powerful predictors and connectomic predictors become more powerful predictors as the disease progresses. This complements previous

work that suggests that atrophy in Parkinson's is the result of network-mediated spread of alpha-synuclein (Luk et al., 2012, Henderson et al. 2019, Zheng et al., 2019). Furthermore, we find a similar trend of increased connectomic influence for severity of psychotic symptoms: namely, schizotypy and schizophrenia. Although schizotypy is not an earlier stage of schizophrenia (indeed, schizotypy is not a disorder per se but rather a multidimensional continuum of traits related to psychosis), individuals with schizotypy exhibit similar, albeit attenuated, characteristics as schizophrenia patients (Kirschner et al., 2021). We find that the connectomic influence is considerably greater in schizophrenia compared to schizotypy, which may suggest that the structural network gradually becomes more implicated in disease progression. One key factor that we were not able to study in more depth is that of sex. Since ENIGMA datasets are all sex-corrected, we are unable to make conclusions about how molecular and connectomic contributions may differ between the sexes. Multiple disorders show well-established sex-differences, including schizophrenia (Abel et al., 2010), autism (Werling & Geschwind, 2013), and depression (Altemus et al., 2014). Designing effective clinical interventions will require more nuanced studies that consider the many heterogeneities that exist within disease."

Abel, K. M., Drake, R., & Goldstein, J. M. (2010). Sex differences in schizophrenia. *International review of psychiatry*, 22(5), 417-428.

Werling, D. M., & Geschwind, D. H. (2013). Sex differences in autism spectrum disorders. *Current opinion in neurology*, 26(2), 146.

Altemus, M., Sarvaiya, N., & Epperson, C. N. (2014). Sex differences in anxiety and depression clinical perspectives. *Frontiers in neuroendocrinology*, 35(3), 320-330.

2) The extent to which the global hypothesis is refuted by this work depends much on the quality of the global phenotypes. For example, if the noise level is higher in the global phenotypes, then they will be found to be less of a contributor to cortical abnormality even if a "noise free" version of the global phenotypes is actually more predictive. Can the authors somehow assess the noise level in the local -vs- global phenotypes (maybe with a simulation?) Or possibly repeat the construction of the global attributes using a different preprocessing pipeline?

What I mean is that 20 years from now, we'll have better technology for assessing graph properties of brains. How confident are the authors that their global evidence will still hold in 20 years, and is there anything more we can do now to gain confidence?

To gain confidence in the connectomic predictors used in this analysis, and to ensure the conclusions are rooted in robust methodology and data, we have repeated all analyses using an additional dataset that was acquired using different imaging and processing pipelines.

Specifically, we repeated the calculation of connectivity predictors using diffusion data from the Human Connectome Project (N=326 unrelated participants, 145 males). HCP data were acquired on a different scanner model (Siemens Skyra instead of Siemens Medical), using a different diffusion MRI and tractography protocol (probabilistic streamline tractography, 18 b0 images, max b-value=3000s/mm², voxel size=1.25mm³, 270 diffusion directions; previously deterministic streamline tractography, 1 b0 image, max b-value=8000s/mm², voxel size=2.2*2.2*3mm, diffusion-spectrum imaging with 128 diffusion

directions), and structural connectivity matrices were generated using different software (MRtrix3; previously Connecome Mapping Toolkit).

We find that the predictors remain consistent across datasets, as do the results from the dominance analysis in Figure 2. This is encouraging and suggests that the global phenotypes used in the present report are robust and consistent across datasets, acquisition parameters and processing methodologies.

We have added a Supplementary Figure (Figure S10) showing the correspondence of global connectivity predictors across datasets:

“Figure S10. Comparing connectome predictors from two different datasets | Connectome predictors were calculated using a structural network from diffusion-spectrum MRI data from the Lausanne dataset (used in the main text) and diffusion-weighted MRI data from the Human Connectome Project (used in the supplement). Connectome predictors are consistent across datasets, acquisition parameters, and processing methodologies.”

As well as a Supplementary Figure (Figure S11) showing the dominance analysis when HCP data was applied.

a | multilinear models

b | dominance analysis

“Figure S11. Replication using HCP structural networks | The main analysis (corresponding to Figure 2 in the main text) was repeated using diffusion weighted MRI data from the Human Connectome Project (N=326) (Van Essen et al., 2013).”

We also mention this explicitly in the text (“Results” section, “Sensitivity and robustness analyses” subsection, Paragraph #1):

“To ensure the results are not driven by choice of dataset, acquisition parameters and processing methodology, we repeated the analyses using structural and functional networks from the Human Connectome Project (N=326, see *Methods* for details), for which acquisition parameters and processing methodologies differ. The connectomic predictors from the Lausanne dataset used in the main text are highly correlated with the same metrics calculated using HCP data (Figure S10). As a result, the regression models and dominance analyses show consistent results (Figure S11).”

3) This might be a good validation of the methodology: can the authors look at pairwise genetic correlation between the pathologies and conditions they study, and see if the pathologies and conditions that share genetic architecture significantly are also close together in the Figure 3 axis? This seems like a good control (towards gaining the sort of confidence I asked for in the previous point) and if the global attributes are robust they should emerge correlated in traits that have similar genetic etiology.

We thank the Reviewer for this suggestion and agree that this test would be not only convincing but also of great interest to the general field. We unfortunately do not have access to (nor the expertise to analyze) population-level genetic correlations among disorders. However, we do find the idea of comparing genetically similar disorders interesting. In Figure 1 of Lee et al., 2019, the authors show the genetic similarity (based on correlations of SNPs) of eight different psychiatric disorders - 6 of which are included in the present manuscript. Interestingly, the strongest association is between schizophrenia and bipolar disorder, two disorders with similar connectomic R^2 . Lee et al., 2019 also find strong relationships between ADHD, ASD, and MDD - the three disorders that we find have the greatest molecular R^2 . Furthermore, Radonijic et al., 2021 compare ENIGMA-derived structural similarity of neuropsychiatric disorders to their genetic similarity from open GWAS studies and find a positive relationship between the two. In other words, disorder profiles that are more similar to one another tend to also have similar genetic profiles. Although relating these results to our findings can only be done in a qualitative capacity, they present a promising future direction for transdiagnostic, cross-disorder research.

We have expanded on this in the text (see also our response to Reviewer #1 Comment #1 who has a similar point about cognitive phenotypes).

Figure 1a in Lee et al., 2019

Figure 1b in Lee et al., 2019

Figure 3 in Radonjić et al., 2021

(“Discussion” section, Paragraph #7):

“This work considers multimodal molecular and connectomic contributions to disorders but does not make conclusions about two important features of disease: cognitive phenotypes and genetics. An exciting future direction is to explore whether molecular and connectomic contributions to disease can be related to phenotypic or genetic similarity. Lee et al., 2019 compare single-nucleotide polymorphism data across eight psychiatric disorders and find that schizophrenia and bipolar disorder show greatest genetic similarity. This complements our finding that schizophrenia and bipolar disorder have consistent connectomic profiles. Lee et al., 2019 also find a clique among the disorders that we find are best predicted by molecular features: ADHD, autism, and major depressive disorder. On the other hand, a comprehensive battery of cognitive and behavioural tests was not uniformly applied to all the disease groups in the ENIGMA datasets. As a result, robust cross-disorder phenotypic profiles are less well established. Our findings potentially suggest an executive function (anchored by schizophrenia) versus attention (anchored by ADHD and ASD) cognitive axis that separates connectomic versus molecularly informed disorder profiles, but more work is needed to standardize cognitive testing and assess how cognitive/behavioural phenotypes may be related to brain structure. Altogether, future work is necessary to explore how overlapping

genetic and neurocognitive disturbances correspond to molecular and connectomic contributions to disease.”

Lee, P. H., Anttila, V., Won, H., Feng, Y. C. A., Rosenthal, J., Zhu, Z., ... & Burmeister, M. (2019). Genomic relationships, novel loci, and pleiotropic mechanisms across eight psychiatric disorders. *Cell*, 179(7), 1469-1482.

Radonjić, N. V., Hess, J. L., Rovira, P., Andreassen, O., Buitelaar, J. K., Ching, C. R., ... & Faraone, S. V. (2021). Structural brain imaging studies offer clues about the effects of the shared genetic etiology among neuropsychiatric disorders. *Molecular psychiatry*, 26(6), 2101-2110.

Minor points:

4) Some short forms such as pd for Parkinson's and 22q for 22q11.2 deletion are inconsistently used (i.e., sometimes the short form is used othertimes not, which could be confusing).

We have revised the manuscript to use consistent language throughout. In the case of Parkinson's and 22q11.2 deletion syndrome, we use their long-form labels throughout.

5) If it's possible, the title could be changed so that it's more clear that "shape" is a verb, for example "Cross-disorder cortical abnormalities are shaped by ..." I know that's longer, maybe there's another way. As it is, it is a bit hard to parse.

Original:

“Molecular and connectomic vulnerability shape cross-disorder cortical abnormalities”

New:

“Local molecular and global connectomic contributions to cross-disorder cortical abnormalities”

6) On page 3, there is a typo "cortical abnormalitiy" should be "cortical abnormality"

Fixed.

7) For Figure 5: x-axis should be on same scale across all except panel a. Pearson's r in some panels looks like < -1, is that because of jitter? panel b: The authors claim this is a normal distribution, can they provide a ks test? panel g: the top 2 and bottom 2 should likely be switched to match caption. Finally, significant correlation is not a contrast to low correlation. None of these are highly correlated (but, they are indeed all positively correlated).

We have updated Figure 5 according to the Reviewer's comments (Figure 5 shown at the end of the response). Note that the correlations are no longer distance-regressed but instead assessed against the spin-test (in response to Reviewer #2 Comment #4). Specifically,

- The x-axes in panels b-h are all identical in the revised version of the figure.
- All points in the scatter plots are within the [-1, 1] range.
- The Kolmogorov-Smirnov test for normality was significant ($s=0.23$, $p<0.001$) which means disorder similarity is significantly different from a normal distribution. We therefore no longer claim that the distribution is approximately normal.
- The caption for panel g has been fixed to match the figure.

- We agree with the Reviewer that a significant correlation does not mean that the effect is strong. Nonetheless, the conclusion of this section is that brain regions with similar molecular make-up tend to be similarly affected across disorders. The key panels are panels c, e, and f where correlation coefficients are >0.40 which we feel is strong enough evidence to make our conclusion.

8) Typo in caption of Figure 1: "inhibitory".

Fixed.

REVIEWER COMMENTS

Reviewer #1 (Remarks to the Author):

The authors have been very responsive to my prior concerns and addressed them accordingly - I also thank the authors for their collegial response letter. I have only very minor residual comments, numbered according to my original review

1. Although not relevant to the present paper, the lack of detailed phenotypic information that accompanies the ENIGMA data seems a challenge going forward. As a clinician, a binary segregation into patients and controls seems anachronistic and I hope efforts are made to address this, either through imputation on suitable subsamples, or changes to data acquisition practices prospectively.

2. Is there any simple way of making formal inference on the trends shown in Figure S4 and the accompanying text.

10. The simplest way of performing spatial resampling whilst preserving the correlation structure and not inflating the variance of the null distribution is through 2D spatial wavestrapping [1] - it's not clear to me why this relatively simple method has not been adopted into this fledgling field.

Reference:

1. Breakspear, M., Brammer, M. J., Bullmore, E. T., Das, P., & Williams, L. M. (2004). Spatiotemporal wavelet resampling for functional neuroimaging data. *Human brain mapping*, 23(1), 1-25.

Reviewer #2 (Remarks to the Author):

I want to congratulate the authors for such an elaborated rebuttal letter. They have addressed all my major concerns with additional analysis, and they have clarified the minor points:

I agree with the authors that removing uncorrected p-values from the dominance analyses is a good approach since the aim of the analysis is to compare input variables. Otherwise, it could have been misleading.

Authors have demonstrated that epicentre likelihood is not related to Euclidean distance.

Statistical tests have been re-run using the spin method.

Some of the concepts and methodological aspects have been clarified.

In summary, my methodological concerns have been completely addressed and the resulting manuscript (together with the SM) will be of major interest to the neuroscience community.

Reviewer #3 (Remarks to the Author):

The authors have provided a revised version of the manuscript "Local molecular and global connectomic contributions to cross-disorder cortical abnormalities." The main conclusion of this paper is that local phenotypes in the brain (such as gene expression and myelination) are better predictors of disease than global phenotypes (such as brain networks). This work is of interest to the neuroscience community.

The authors have made many improvements to the manuscript since the last version, and answered reviewer questions in detail. The main improvements are:

- 1) More comparison of results across the diseases examined.
- 2) More analysis of the correlation structure of the predictors.
- 3) Replication of results on an independent dataset: the Human Connectome Project (HCP). This involved a modern MRI technology, and provided validation of the main conclusion.

This follow up work is high quality, and I don't have further comments. I did recommend a sex stratified analysis in my previous review, which was not done. But the authors have now discussed sex stratification in the discussion, and this may be an avenue of future work.

Dear Reviewer,

Thank you for the conditional acceptance of our revised submission. We feel the revision process was fair and considerably improved the manuscript. Here we respond to the remaining Reviewer comments, and make minor changes to the manuscript. Reviewer comments are in **bold font** and our responses are in regular font.

Reviewer #1 (Remarks to the Author):

The authors have been very responsive to my prior concerns and addressed them accordingly - I also thank the authors for their collegial response letter. I have only very minor residual comments, numbered according to my original review

1. Although not relevant to the present paper, the lack of detailed phenotypic information that accompanies the ENIGMA data seems a challenge going forward. As a clinician, a binary segregation into patients and controls seems anachronistic and I hope efforts are made to address this, either through imputation on suitable subsamples, or changes to data acquisition practices prospectively.

We fully agree with the Reviewer that the lack of phenotypic information is a limitation of the ENIGMA datasets. The primary challenge lies in harmonizing cognitive and clinical measures. The ENIGMA consortium relies on collected data, and has well-established guidelines for harmonizing the processing and analytic pipelines of neuroimaging and genetic data. However, guidelines for combining phenotypic information across multiple cohorts remain unclear, in part because not all sites collect phenotypic data, but more importantly because different groups have different assessment methods. One option would be to convert site-specific cognitive/behavioural scores to a standardized score based on population mean and standard deviation. Encouragingly, new ENIGMA Working Groups (for example the Brain Injury Working Group) are putting together recommended practices for harmonizing phenotypic data (Wilde et al., 2021, psyarxiv). These protocols will hopefully be adopted across all the Working Groups in the near future.

Wilde, E. A., Dennis, E. L., & Tate, D. F. (2021). The ENIGMA brain injury working group: Approach, challenges, and potential benefits. *Brain imaging and behavior*, 15(2), 465-474.

2. Is there any simple way of making formal inference on the trends shown in Figure S4 and the accompanying text.

We now conduct F-tests between two regression models with the same number of input variables to statistically compare whether one model performs better than another. In the case of the age-stratified analysis, we conduct the F-test between the adulthood stage and the adolescent stage (or pediatric, when adolescence isn't available). We find significant increases (i.e. $F > F_{crit}$) for ADHD and depression using the molecular predictor set. For the Parkinson's disease severity analysis, we do not find that the increase/decrease in model fit is statistically significant, so we mention this directly in the text and in the figure caption.

("Results" section, "Local and global contributions to disorder-specific cortical morphology" subsection, Paragraph #4):

“First, we tracked the model fit (R^2_{adj}) of regression models that fit molecular/connectomic features to pediatric, adolescent, and adult cortical abnormality profiles for the four available disorders with this data (ADHD, bipolar disorder, depression, and OCD; Supplementary Fig. 4a). We find that model fit is greatest in adulthood, except for OCD which shows little change for connectomic predictors and a lower model fit in adulthood for molecular predictors. Model fit significantly improves when molecular features are used to predict cortical abnormality patterns of ADHD and depression ($F > F_{critical}$, one-sided). Next, focusing on disease severity, we show how model fit changes across four levels of Parkinson’s disease severity (Hoehn and Yahr (HY) stages (Hoehn & Yahr, 1967); Supplementary Fig. 4b). Interestingly, we find that from stage HY2, molecular predictors perform worse with disease severity whereas connectomic predictors perform better (although note the changes in model fit are not statistically significant), supporting the notion that Parkinson’s pathology is influenced by the spread of misfolded proteins on the structural connectome (Luk et al., 2012, Henderson et al., 2019, Zheng et al., 2019). Altogether, these analyses provide a more nuanced and transdiagnostic representation of molecular and connectomic contributions to cortical disorder vulnerability.”

10. The simplest way of performing spatial resampling whilst preserving the correlation structure and not inflating the variance of the null distribution is through 2D spatial wavestrapping [1] - it's not clear to me why this relatively simple method has not been adopted into this fledgling field.

Spatial wavestrapping is theoretically a promising avenue for resampling the brain surface. However, 2D spatial wavestrapping is designed for two dimensional data like fMRI slices. This method would have to be extended to 3D surface data before it can be used in a similar manner as the spin test. A surface-version of the 2D spatial wavestrapping method would no doubt be of great use to the neuroimaging community but is beyond the scope of this study.

We mention alternative nulls in the text (“Methods” section, “Null models” subsection, Paragraph #1):

“Other methods for constructing spatial null models exist, such as generative models (Burt et al., 2020) and 2D spatial wavestrapping (Breakspear et al., 2004).”

Reference:

1. Breakspear, M., Brammer, M. J., Bullmore, E. T., Das, P., & Williams, L. M. (2004). Spatiotemporal wavelet resampling for functional neuroimaging data. *Human brain mapping*, 23(1), 1-25.